# Transposons contribute to the acquisition of cell type-specific cis-elements in the brain

Kotaro Sekine [1,2], Masahiro Onoguchi [1,2✉] & Michiaki Hamada [1,2,3✉]

Mammalian brains have evolved in stages over a long history to acquire higher functions. Recently, several transposable element (TE) families have been shown to evolve into cis-regulatory elements of brain-specific genes. However, it is not fully understood how TEs are important for gene regulatory networks. Here, we performed a single-cell level analysis using public data of scATAC-seq to discover TE-derived cis-elements that are important for specific cell types. Our results suggest that DNA elements derived from TEs, MER130 and MamRep434, can function as transcription factor-binding sites based on their internal motifs for Neurod2 and Lhx2, respectively, especially in glutamatergic neuronal progenitors. Furthermore, MER130- and MamRep434-derived cis-elements were amplified in the ancestors of Amniota and Eutheria, respectively. These results suggest that the acquisition of cis-elements with TEs occurred in different stages during evolution and may contribute to the acquisition of different functions or morphologies in the brain.

[1] Graduate School of Advanced Science and Engineering, Waseda University, Tokyo, Japan. [2] Computational Bio Big-Data Open Innovation Laboratory (CBBD-OIL), National Institute of Advanced Industrial Science and Technology (AIST), Tokyo, Japan. [3] Graduate School of Medicine, Nippon Medical School, Tokyo, Japan. ✉email: m.onoguchi@aoni.waseda.jp; mhamada@waseda.jp

In the mammalian brain, neurons are mainly responsible for the transmission and processing of information in the brain, and there are various spatiotemporal subtypes, including layer-specific excitatory neurons in the cerebral cortex[1]. These neurons are differentiated from neural progenitors during embryonic and adult neurogenesis[1–4]. For adult neurogenesis, neural stem cells (NSCs) are first committed to proliferative neural progenitor cells (NPCs) and then become neuronal progenitor cells and neuroblasts[3,4]. These cells then differentiate into glutamatergic excitatory neurons or GABAergic interneurons under the control of various transcription factors at different differentiation states[3,4]. In addition, transcription factors such as Neurod2 and Lhx2 play various roles in neurogenesis. For example, Neurod2 induces neuronal differentiation in NPCs and contributes to neuronal maturation[5–7]. Lhx2 has multiple spatiotemporal roles by regulating the proliferation and differentiation of NPCs, as well as area patterning and axonal wiring in postmitotic neurons[8]. Recent studies have shown that while adult neurogenesis occurs mainly in the subgranular zone (SGZ) or the subventricular zone (SVZ), it also could occur in other regions, such as the cortex[9]. In addition, environmental factors, such as stress and injury, are involved in regulating the generation and survival of newborn neurons[9–11].

Recent single-cell analyses have shown that spatiotemporally diverse neuronal subtypes have distinct genomic profiles[12,13]. Among them, pseudo-time analysis can characterize the different stages of neurogenesis in single-cell levels. For example, pseudo-time analysis with single-cell RNA-seq (scRNA-seq) revealed gene expression profiles during glutamatergic neurogenesis in the adult dorsal ventricular-subventricular zone (V-SVZ) and GABAergic neurogenesis in the adult ventral V-SVZ[14]. As for epigenetic analysis, some studies have reported dynamic changes in chromatin accessibility profiles during neurogenesis using single-cell assays for transposase accessibility by sequencing (scATAC-seq) in mammals[15,16]. scATAC-seq analyses in the brain have also revealed transcription factor motifs accessible in a cell-type-specific manner in spatiotemporally diverse neuronal

subtypes (such as excitatory/inhibitory neurons or neural progenitors)[17–19]. However, it remains to be elucidated how cell-type-specific regulatory mechanisms are acquired.

Previous and recent studies have proposed that transposable elements (TEs) may be important in understanding gene regulatory mechanisms and their origins[20–22]. TEs are DNA sequences that can change their position within a genome and occupy approximately half of the mammalian genome[23,24]. TEs are subdivided according to their evolutionary origin, metastasis mechanism, and sequence characteristics. For example, in the mouse genome, there are approximately 52 families and 790 subfamilies[23]. Specific types of TEs can function as gene regulatory regions by serving as the origin of transcription factor (TF) binding sites[25–27]. Most TEs that function as transcription factor-binding sites also contain transcription factor-binding motifs, suggesting that transcription factors bind to TEs in a sequence-dependent manner[25]. For example, recent studies reported that MER130 TEs function as cortical-specific enhancers based on their internal motifs[26,27]. TEs may also be important factors for the acquisition of novel cis-elements during evolution. Some groups suggested that TEs are involved in acquiring cis-elements, including transcription factor-binding sites that drive morphological evolution in mammals[22,28–30]. For example, recent studies have shown that the acquisition of AmnSINE1-derived enhancers in mammalian ancestors may lead to the acquisition of mammalian-specific brain formation[31,32]. These studies suggest that TEs may contribute to the acquisition of cell-type-specific cis-elements by inducing the binding sites of transcription factors. However, the relationship between TE-derived DNA elements and transcription factor binding at the single-cell level remains to be elucidated.

In this study, we verified the hypothesis that TEs contribute to the acquisition of transcription factor-binding motifs that are important for cell-type specificity (Fig. 1a). To clarify whether TE-derived cis-elements serve as binding sites for cell-type-specific transcription factors, we used public scATAC-seq data from the adult mouse (P56) prefrontal cortex[33]. Specifically, using

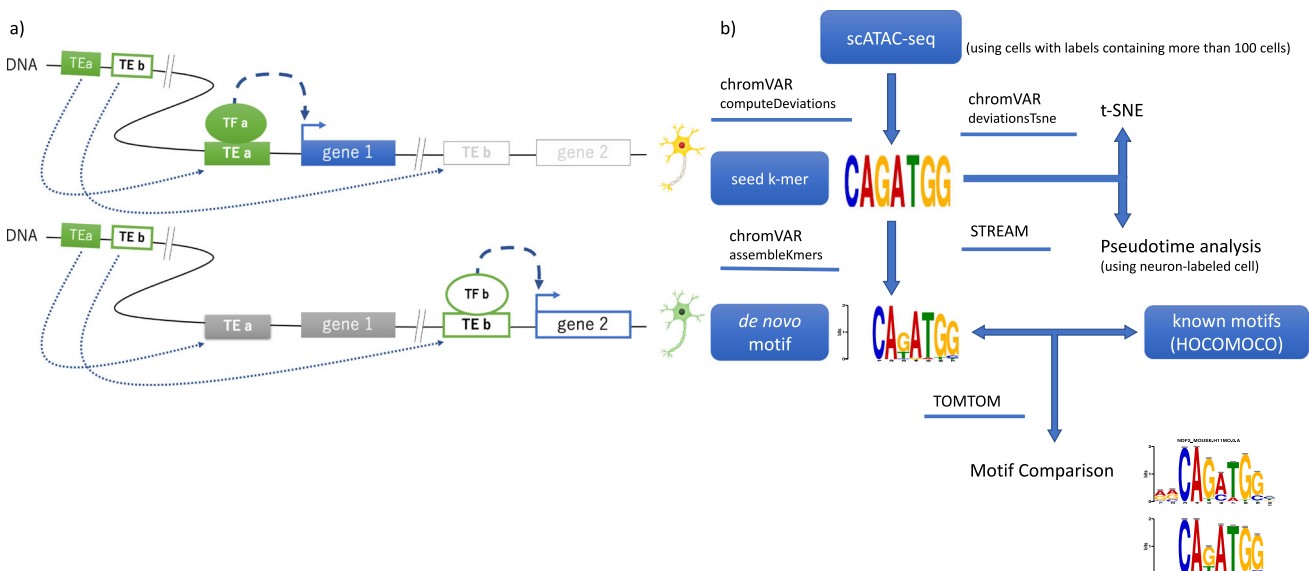

**Fig. 1 Hypothesis that transposable elements (TEs) contribute to the acquisition of transcription factor (TF)-binding motifs that are important for cell-type specificity. a** A specific type of TE acquired during evolution serves as binding sites of transcription factors and induces cell-type-specific gene expression. In the cell type shown in the upper figure, TEs that differ from those in other cell types shown in the lower figure function as cis-elements and form a gene regulatory mechanism specific to the cell type. The solid, dashed, and dotted arrows indicate the gene expression, gene regulation, and TE insertion, respectively. **b** Analysis flow of de novo motifs using scATAC-seq in this study. The data status is enclosed in a rectangle, and the tools we used are underlined.

scATAC-seq data with labels (defined by Cusanovich et al.[33]), we first detected transcription factor motifs accessible in a cell-type-specific manner (Fig. 1b). Then, we searched for TE subfamilies that function as cell-type-specific binding sites of transcription factors. Finally, by estimating the time of acquisition of TE-derived cis-elements, we showed the role of TEs in the acquisition of cell-type-specific brain functions.

## Results

**De novo motifs with high variability in chromatin accessibility across cells are similar to known binding motifs of neural differentiation-related transcription factors.** To discover accessible DNA motifs that are important for cell-type specificity in the mouse adult prefrontal cortex (P56), we first investigated whether cell types are characterized by k-mer chromatin accessibility profiles with scATAC-seq (accessibility across peaks sharing the k-mer), using cells with known cell-type labels, each containing more than 100 cells (labels were defined by Cusanovich et al.[33]). First, we computed the bias-corrected accessibility deviations for each 7-mer with chromVAR[34] (see "Methods"). Visualization of the similarity among cells with t-distributed stochastic neighbor embedding (t-SNE) showed that cells were characterized as upper/lower layer excitatory neurons, inhibitory neurons, and glial cells (Fig. 2a, Supplementary Fig. 1). This suggests that the k-mer chromatin accessibility profiles differ greatly among various types of neurons and glial cells.

Then, using the k-mer accessibility profiles, we generated de novo motifs associated with high variability in chromatin accessibility across the cells with chromVAR's assembleKmers function[34]. Thus, we obtained four de novo motifs (the threshold of variability score = 2.0). The variability scores of the seed k-mers in chromVAR[34] were as follows: de novo motif 1: CAGATGG was 3.7, de novo motif 2: CACCCAC was 3.0, de novo motif 3: CTAATTA was 2.1, and de novo motif 4: ACAGCTG was 2.1.

To identify the cell types for which the seed k-mers of the four de novo motifs with high accessibility variation across the cells are accessible, we visualized the k-mer deviation z-scores calculated with chromVAR[34] for each cell on t-SNE (Fig. 2b, Supplementary Fig. 2). The results showed that each seed k-mer of the de novo motif 1 and de novo motif 2 were accessible mainly in excitatory neurons. In contrast, de novo motif 3 and de novo motif 4 were accessible across multiple cell types but were relatively enriched in excitatory neurons and astrocytes for de novo motif 3 and interneurons and inhibitory neurons for de novo motif 4. In excitatory neurons, the seed k-mers of de novo motifs did not precisely correspond to any known labels of neuronal subtype, and they were accessible in a part of each neuronal subtype of the excitatory neurons. These findings suggest that these de novo motifs may regulate some specific subpopulation of excitatory neurons.

Then, to estimate which transcription factors contribute to the accessibility of each de novo motif, we compared the de novo motifs with known motifs of transcription factors (see "Methods"). We obtained several known transcription factor candidates whose motifs are significantly ($q < 0.05$ using TOMTOM[35]) similar to each of the de novo motifs (Fig. 2c, Supplementary Fig. 3). Some of these candidate transcription factors have been reported to play important roles in neuronal differentiation[2,7,8], suggesting that transcription factor motifs related to the regulation of neuronal differentiation are important for open chromatin profiles that characterize cell types. For example, de novo motif 1 is similar to the transcription factor-binding motif shared by the neural-specific basic-helix-loop-helix (bHLH) transcription factors, such as Neurod2, Neurog2, and Neurod1[36].

This suggests that this motif is implicated in the differentiation of glutamatergic neurons[3,7] (Supplementary Fig. 3).

To check to what extent the candidate transcription factor genes are actively transcribed in the cell types where the de novo motifs are accessible, we examined their RNA expression levels using public scRNA-seq data of the adult mouse cortex[13] and gene activity levels using scATAC-seq data[33] (see "Methods"). We confirmed that specific transcription factors of the candidates that regulate neural differentiation and/or neuronal activity are highly expressed in each cell population[7,8,37–40] (Fig. 2d, Supplementary Fig. 4). In the cells where de novo motif 1 was highly accessible (that is, excitatory neurons), the results showed that *Neurod2* was expressed the most compared to that of other candidates of transcription factors, including *Neurog2*, *Neurod1*, *Atoh1*, and *Olig2*. Therefore, Neurod2 was assumed to be the factor that contributed most to de novo motif 1 accessibility. Similarly, Egr3 for de novo motif 2 (accessible in excitatory neurons), Lhx2 for de novo motif 3 (accessible in excitatory neurons and astrocytes), and Tcf4 for de novo motif 4 (accessible in interneurons and inhibitory neurons) were estimated to be transcription factors that contribute the most to the de novo motif accessibility (Fig. 2c). Notably, de novo motif 3, which corresponds to putative Lhx2 binding sites, was highly accessible across a part of excitatory neurons and astrocytes (Fig. 2b, d). This suggests that Lhx2 regulation is important for not only neurons but also astrocytes, consistent with the results of a recent report showing that Lhx2 plays an essential role in astrocyte maturity through transcriptional and chromatin regulation[41].

**Each Neurod2 and Lhx2-like motif is accessible in a putative neuronal progenitor cell population in the adult brain.** Because we found that transcription factors crucial for neuronal differentiation might regulate the de novo motifs of highly variable accessibility (Fig. 2), we further characterized subpopulations that showed open chromatin profiles in the de novo motifs in neuron-labeled cells along the differentiation axis. We performed pseudo-time analysis on the neuron-labeled cells using the k-mer accessibility profiles. We ordered the cells along with the neuronal differentiation state using STREAM[42] (Fig. 3). We found that the excitatory neurons branched into two branches (S2, S1) and (S0, S1): the branch (S0, S1) was the cell population in which excitatory neuron-labeled cells accounted for 83% of the total (3071/3678 cells), and the branch (S2, S1) was the cell population where excitatory neuron-labeled cells accounted for 97% of the total (414/429 cells) (Fig. 3a, Supplementary Fig. 5). Among them, in branch (S2, S1), layer VI excitatory neurons (Ex._neurons_CThPN: corticothalamic projection excitatory neurons) accounted for 65% of the total (278/429 cells). The branch (S3, S1) was the cell population where inhibitory neuron-labeled cells accounted for 56% of the total (81/145 cells).

To understand the developmental processes that characterize each of the branches, we focused on the marker gene activities calculated with ATAC peaks near the genes (see "Methods"). The marker genes of glutamatergic neuronal progenitor cells and neuroblasts, such as *Neurog2* and *Neurod1*[3,43], were more active in branch (S2, S1), whereas genes related to maturation, such as *Mef2c*[44], were more active in branch (S0, S1) (Supplementary Fig. 7a, b). In addition, *Dlx6* and *Gad2* genes, which are related to the function of GABAergic neurons[45], were active in branch (S3, S1) (Supplementary Fig. 7c). These results suggest that branch (S2, S1) may be a younger neuronal population, such as neuronal progenitors or neuroblasts, while branch (S0, S1) may be a mature neuronal population, and branch (S3, S1) may be GABAergic inhibitory neurons. We also performed Gene Ontology (GO)

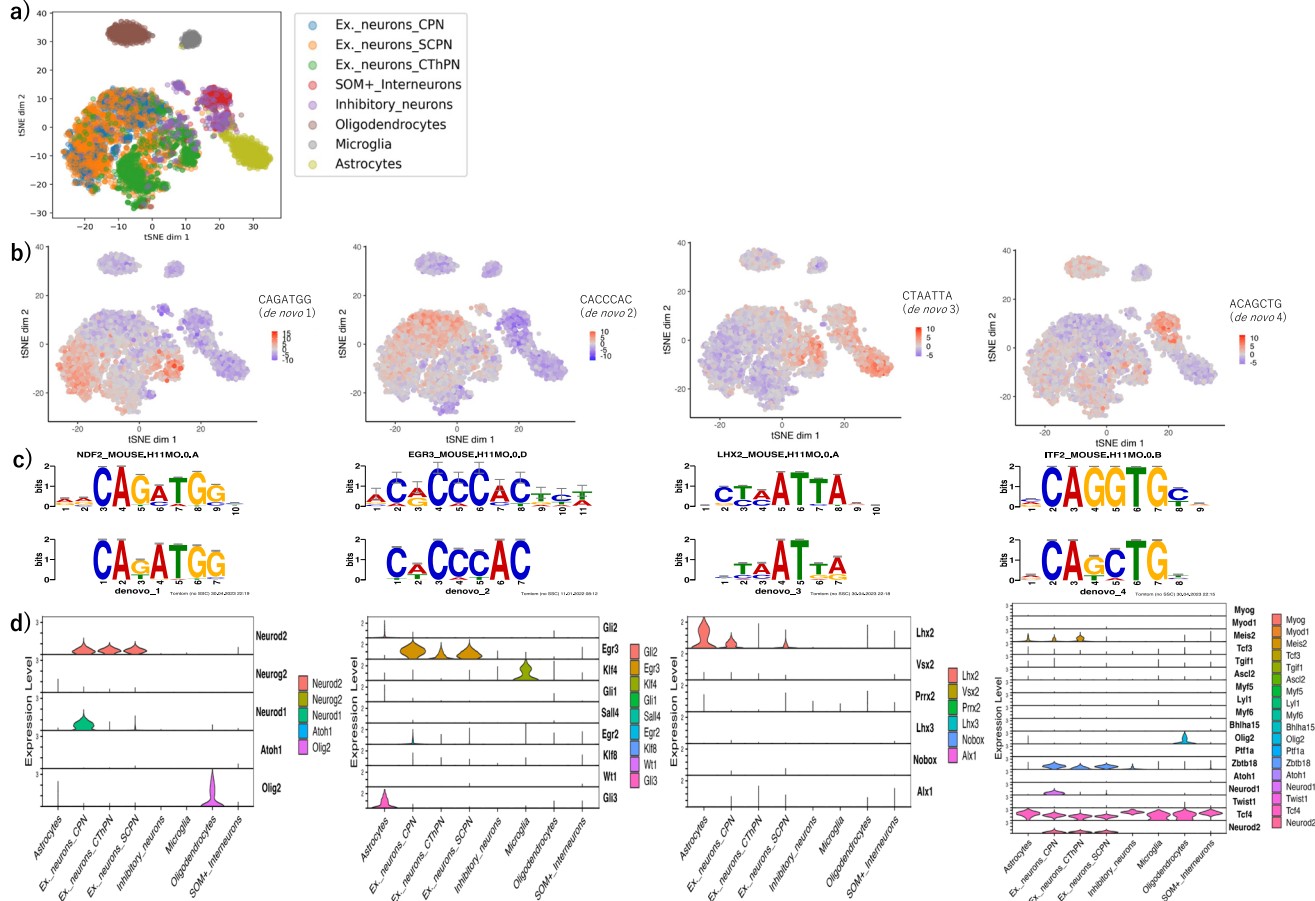

**Fig. 2 De novo motifs with high variability in chromatin accessibility across the cells are similar to the neural development-related transcription factor motifs. a** Visualization of cell similarity using t-distributed stochastic neighbor embedding (t-SNE) based on 7-mer chromatin accessibility. The colors indicate the labels defined in the dataset of the original paper[34] (Ex._neurons_CPN: callosal projection excitatory neurons; Ex._neurons_SCPN: subcerebral projection excitatory neurons; Ex._neurons_CThPN: corticothalamic projection excitatory neurons; SOM+_Interneurons: somatostatin positive interneurons). **b** Visualization of the accessibility of the seed k-mer used to generate the de novo motif with t-SNE. Each cell is colored by deviation z-scores (de novo motif 1: CAGATGG; de novo motif 2: CACCCAC; de novo motif 3: CTAATTA; de novo motif 4: ACAGCTG). **c** Known motifs of transcription factors that function in the brain, which are similar ($q < 0.05$) to de novo motifs. The upper figures in each panel show the known motifs obtained from the HOCOMOCO database[71] that are similar to the de novo motifs, and the bottom figures show the corresponding de novo motifs which were generated based on the k-mers with large accessibility variation across the cells in this study (de novo motif 1: Neurod2 (NDF2); de novo motif 2: Egr3; de novo motif 3: Lhx2; de novo motif 4: Tcf4 (ITF2)). In each panel, the similarity between motifs was examined using TOMTOM[35]. **d** Expression levels of candidate transcription factors with motifs similar to de novo motif 1–4. Each panel shows the expression levels of scRNA-seq data based on the cell-type labels transferred from scATAC-seq data with Seurat[73].

analysis on the lists of differentially activated genes ($\log_2$ fold >2; $q < 0.05$: Mann–Whitney $U$ test with Benjamini–Hochberg correction) between branches (S2, S1) and (S0, S1). The results showed that transport, stress response, and blood circulation were the top enrichment terms for genes that were highly active in branch (S2, S1) (Supplementary Fig. 8). These results suggest that branch (S2, S1) may be a population committed as neuronal progenitors or neuroblasts that are activated in a stress-dependent manner and are involved in tissue repair.

To ask whether de novo motifs were specifically enriched in some branches, we calculated the motif accessibility using chromVAR[34] and compared them between the branches. We found that the seed k-mers of de novo motif 1 (Neurod2-like motif) and de novo motif 3 (Lhx2-like motif) were significantly more accessible in the branch (S2, S1), to which Ex._neurons_CThPN-labeled cells mainly belong, compared to the branch (S0, S1) ($q < 0.01$ for neuron-labeled and Ex._neurons_CThPN-labeled cells, respectively: Mann–Whitney $U$ test with Benjamini–Hochberg correction for bias-corrected accessibility

deviations) (Fig. 3c–f and Supplementary Fig. 6). These results suggest that de novo motifs 1 and 3 function in specific differentiation states of excitatory neurons, which are considered neuronal progenitors or neuroblasts. Hereafter, we refer to this population as the putative neuronal progenitor cell population. This is consistent with studies that have shown that Neurod2 (candidate transcription factor of de novo motif 1) and Lhx2 (candidate transcription factor of de novo motif 3) are involved in the development of glutamatergic neurons[46–48].

**Specific TE subfamilies are enriched in the accessible de novo motifs and transcription factor-binding sites.** Recent studies have shown that TEs could contribute to the regulatory DNA cis-elements in the brain[26,27,30–32]. To ask to what extent TEs play essential roles as cis-elements, we compared the percentage of TEs that overlap with the open chromatin regions among cell types using binarized (if the fragment was present =1, otherwise 0) peak-by-cell chromatin accessibility matrix (see "Methods"). A comparison of the percentage of ATAC peaks that intersect TEs

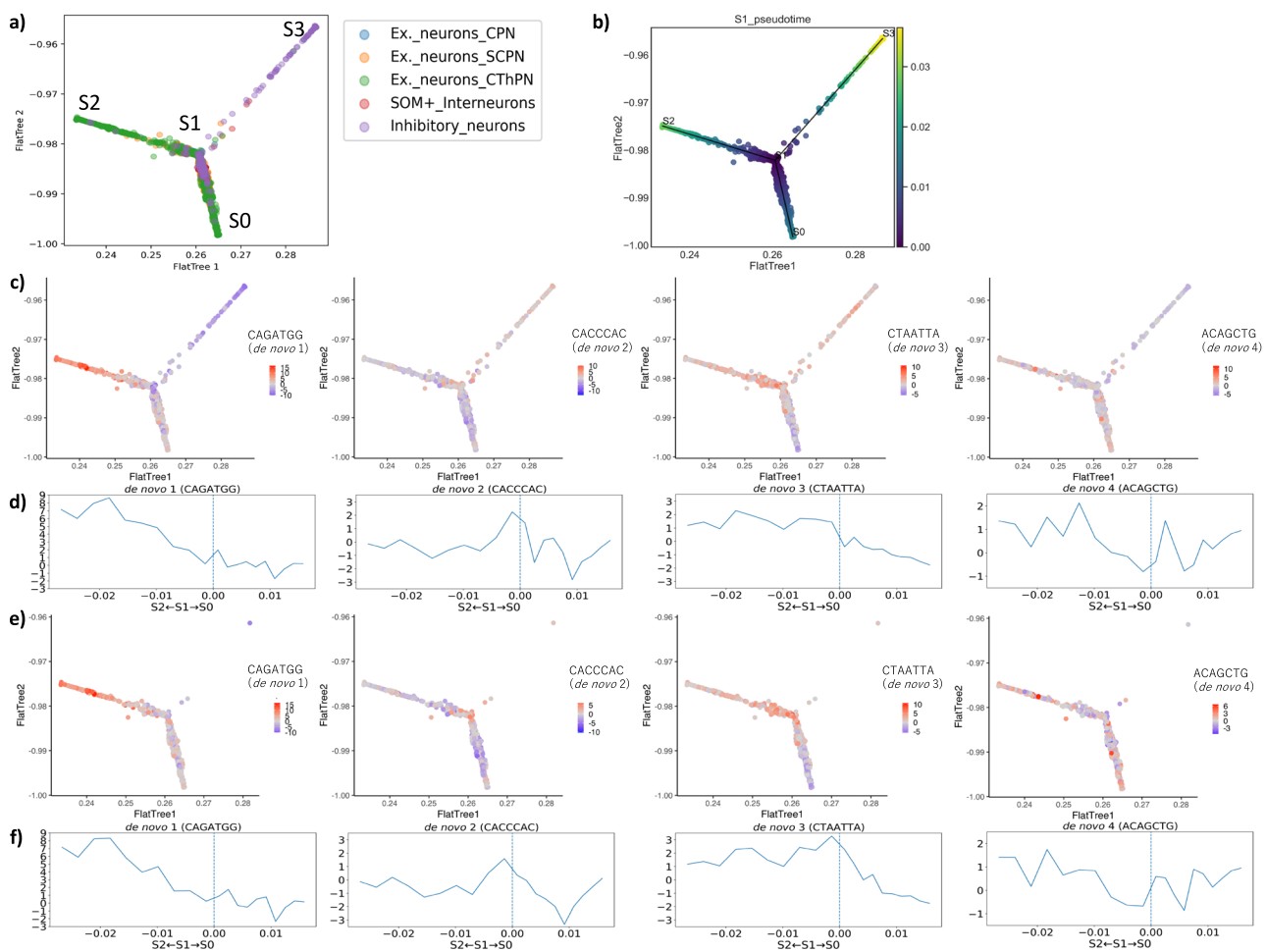

**Fig. 3 Neurod2 and Lhx2-like de novo motifs are accessible in specific differentiation states of excitatory neurons.** Each plot shows the putative cellular differentiation trajectories based on the k-mer profile with STREAM[42]. **a** Points are colored with known cell-type labels. **b** Points are colored by the pseudo-time from node S1. **c**, **e** Points are colored by deviation $z$-scores for seed k-mers of de novo motifs (de novo motif 1: CAGATGG; de novo motif 2: CACCCAC; de novo motif 3: CTAATTA; de novo motif 4: ACAGCTG). For the plots of deviation $z$-scores, cell-type labels are limited to **c** neurons, **e** Ex._neurons_CThPN. **d**, **f** Comparison of the deviation $z$-scores for seed k-mers of de novo motifs between branches (S2, S1) and (S0, S1). The graph shows the average seed k-mer accessibility of de novo motifs for each segment when the pseudo-time of each branch is equally divided into 10 segments. The $x$-axis shows the pseudo-time from the branching point S1 (0, indicated with a blue vertical dotted line) to the nodes S2 ($-0.027$) or S0 (0.016). The $y$-axis shows the average seed k-mer deviation $z$-score for each segment. For the plots of deviation $z$-scores, cell-type labels are limited to **d** neurons, **f** Ex._neurons_CThPN.

among cell types showed that TEs tended to be open chromatin significantly in excitatory neurons compared to the other cell types, such as inhibitory neurons or glial cells (Fig. 4a, Supplementary Data). To account for the differences in the number of cells of each cell type that can affect the quality of pseudo-bulk ATAC peaks, we subsampled the same number of cells for each cell type and compared the proportion of the peaks overlapping with TEs. We obtained a consistent result with data that used the whole cell population and concluded that TEs were significantly more accessible in the excitatory neurons than in other cell types ($q < 0.01$: Fisher's exact test with the Benjamini–Hochberg correction; Supplementary Table 1). In addition, a comparison between the pseudotemporal branches obtained in Fig. 3 showed that TEs tended to be open chromatin in branch (S2, S1) and branch (S0, S1), which was related to the differentiation process of excitatory neurons (Fig. 4b, Supplementary Data). These results suggest that TEs act as cis-elements more in excitatory neurons.

To investigate whether these TE-derived cis-elements function as enhancers or promoters, we focused on histone modifications using chromatin immunoprecipitation coupled with high-throughput sequencing (ChIP-seq) data of cortical tissues. From the ChIP-Atlas database[49], we prepared ChIP peak sets of H3K4me1 and H3K4me3 that are known as enhancer and promoter labels, respectively[50]. The peaks of H3K4me1 and H3K4me3 overlapped 15% (24530/163902) and 18% (29400/163902) of the number of pseudo-bulk ATAC peaks (obtained by merging all of the scATAC-seq peaks, see "Methods"), respectively. Then, we compared whether bulk ATAC peaks tended to overlap with H3K4me1 or H3K4me3 ChIP-seq peaks between TE-derived ATAC peaks and non-TE-derived ATAC peaks. As a result, the TE-derived ATAC peaks tended to overlap significantly with H3K4me1 when compared to non-TE-derived peaks ($p < 0.01$: Fisher's exact test; Table 1). In addition, the TE-derived accessible de novo motifs 1–3 also tended to overlap with H3K4me1 ($p < 0.01$: Fisher's exact test; Supplementary Table 2), suggesting that TE-derived cis-elements function more as enhancers than promoters.

Notably, using TE class labels in Repbase[51], we discovered that the class distribution of TEs overlapping with open chromatin

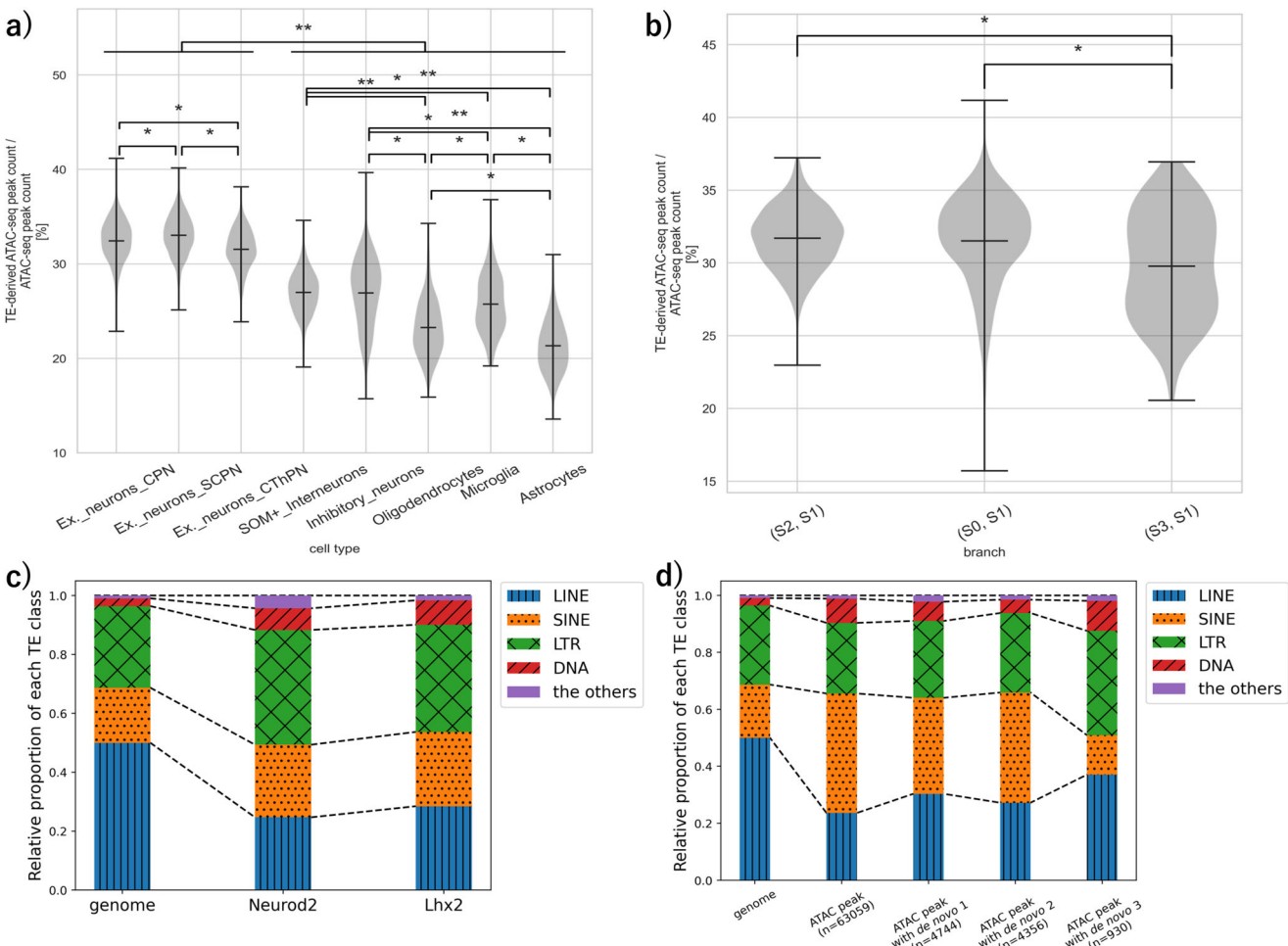

**Fig. 4 TEs tend to be open chromatin in excitatory neurons and specific TEs are enriched in the open chromatin sites. a, b** Violin plots of the proportion of TE-derived ATAC-seq peaks. Each violin plot shows the distribution of the percentage of ATAC peaks that overlap with TEs for each cell belonging to the cell type. The vertical axis shows the percentage of ATAC peaks overlapping with TEs and the horizontal axis shows each cell population defined by Cusanovich et al.[33] (**a**) and obtained from the pseudo-time analysis in Fig. 3b. In each panel, the percentage of ATAC peaks overlapping with TEs was significantly different among each group (Kruskal–Wallis tests, $p < 0.01$). Marks indicate significant differences between the two groups (*$q < 10^{-5}$, **$q < 10^{-50}$: Mann–Whitney's $U$ test with Benjamini–Hochberg correction). In panel (**a**), significant differences were found between excitatory neurons and other cell types at $q < 10^{-50}$ for any combination. **c, d** Distribution of classes of the TE that resides in transcription factor-binding sites (**c**) or ATAC-seq peaks (**d**). The first column shows the relative proportions of each class of TEs in the whole mouse genome as a control. **c** The bar graph shows the relative proportion of ChIP peaks that overlap with each TE class. The percentage of TEs in the ChIP peaks was 5.2% for Neurod2 (63777/1236918 (bp)) and 16% for Lhx2 (695900/4285911 (bp)). **d** The bar graph shows the relative proportion of the number of bulk ATAC peaks ($n$) that overlap with each TE class. For these ATAC peaks, the cases in which de novo motifs 1–3 in the ATAC peaks overlap with the TEs are shown separately. The total number of ATAC peaks with TE regions was 38% (63059/163902). Among the ATAC peaks with motifs, the percentage of the number of peaks with TE-derived motifs was 17% (4744/28382) for de novo motif 1, 18% (4356/24461) for de novo motif 2, and 12% (930/8009) for de novo motif 3.

**Table 1 TE-derived cis-elements tend to function as enhancers.**

|  | H3K4me1+/H3K4me3− | H3K4me1−/H3K4me3+ |
|---|---|---|
| TEs | 7449 | 3118 |
| Non-TE | 12,201 | 21,402 |

Each number indicates the number of bulk ATAC peaks (obtained in "Data sources") that overlap with the corresponding genomic regions. Each row indicates the TE and non-TE regions. For each column, "+" indicates that the histone modification ChIP peak is present, and "−" indicates that the histone modification ChIP peak is absent.

regions was different from their distribution in the genome (Fig. 4c, d, Supplementary Data). For example, the proportion of DNA-class TEs in the accessible de novo motif 1 (Neurod2-like motif) or de novo motif 3 (Lhx2-like motif) was more than twice the proportion of DNA-class TEs in the genome. This suggests that TEs are not randomly residing in cis-elements, and certain types of TEs may function as cis-elements. Thus, we wondered if specific TE subfamilies contribute to the cis-elements essential for regulating specific cell types. Based on the results that de novo motif 1 (Neurod2-like motif) and de novo motif 3 (Lhx2-like motif) were accessible in putative neuronal progenitor cells, we searched for TE subfamilies enriched in binding sites of the

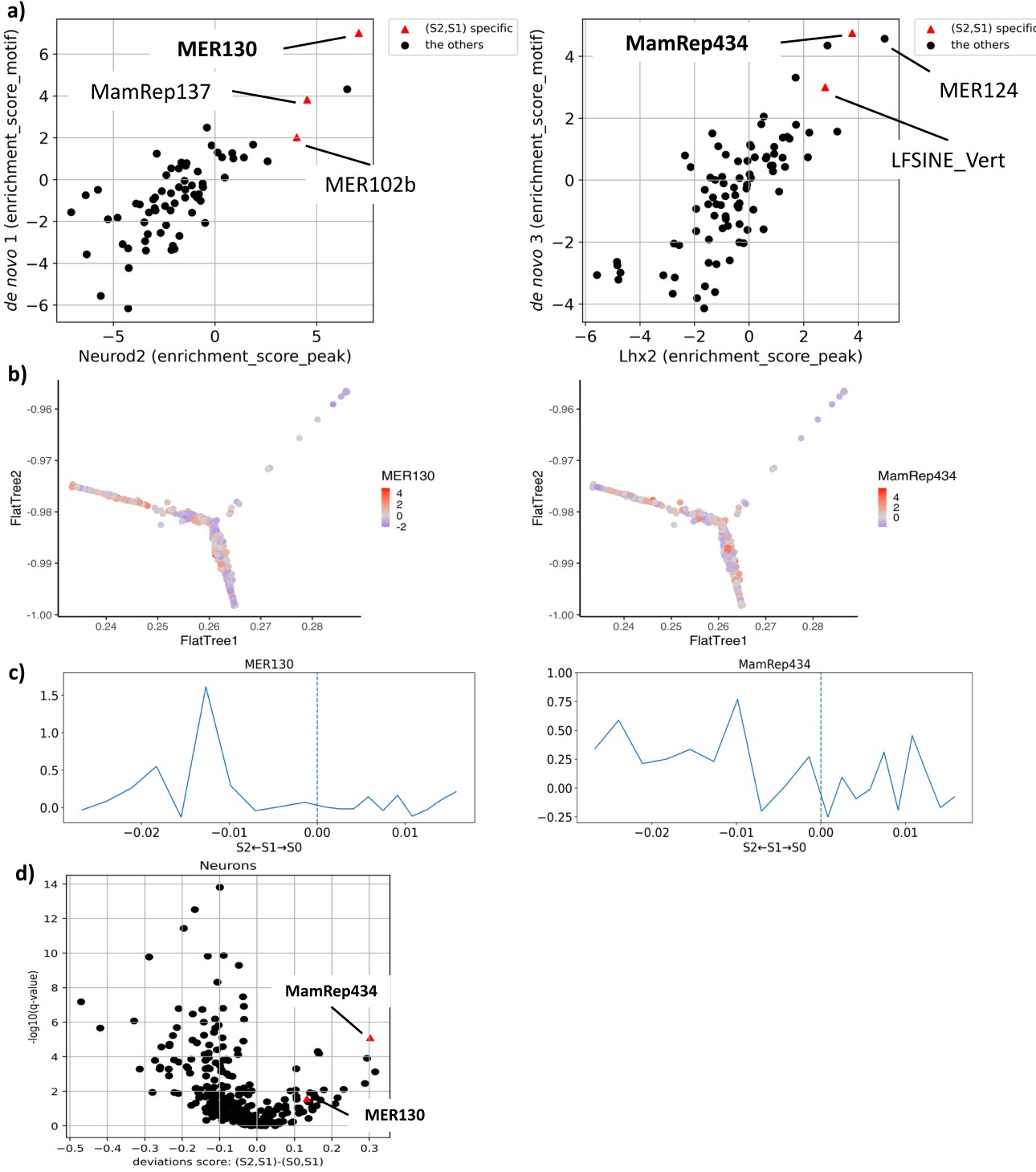

transcription factors (Neurod2 and Lhx2) important for the regulation of this population. We evaluated the enrichment scores of accessible motifs (ATAC peaks with de novo motifs 1 and 3) and transcription factor ChIP peaks to TEs (see "Methods"). As a result, MER130 (unknown class in Repbase[51] but currently considered DNA class[52]) was the most enriched TE subfamily in Neurod2 binding peaks and de novo motif 1 ($q < 0.01$: false discovery rate (FDR)) (Fig. 5a, Supplementary Data). MER124 (DNA class in Repbase[51]) and MamRep434 (DNA class in Repbase[51]) were the most enriched TE subfamilies in Lhx2 binding peaks and de novo motif 3, respectively ($q < 0.01$: FDR) (Fig. 5a, Supplementary Data). To account for technical biases

(GC content and average accessibility) of scATAC-seq peaks, we calculated the $z$-score using the enrichment score of accessible motifs based on the chromVAR's background peak set as a control (see "Methods"). As a result, we confirmed that MER130 and MamRep434 were enriched in de novo motifs 1 and 3, respectively ($z$-score > 2; Supplementary Data). These results suggest that specific TE subfamilies significantly contribute to Neurod2 and Lhx2 binding sites.

**Specific TE subfamilies, including MER130 and MamRep434, are themselves accessible in a putative neuronal progenitor cell population.** Then, to discover TE subfamilies especially important

**Fig. 5 MER130 and MamRep434 are enriched in Neurod2 and Lhx2 binding sites, respectively, and accessible in a putative neuronal progenitor cell population. a** Scatter plots for the TE enrichment scores of transcription factor-binding sites [*x*-axis] (left panel: Neurod2, right panel: Lhx2) and de novo motifs [*y*-axis] (left panel: de novo motif 1, right panel: de novo motif 3). Black circles indicate each TE subfamily, and red triangles indicate TEs significantly accessible in a branch (S2, S1) (*q* < 0.05: Mann–Whitney *U* test with Benjamini–Hochberg correction between the branches (S2, S1) and (S0, S1)). **b** TE accessibility scores of neuron-labeled cells in the putative cellular differentiation trajectory map (left panel: MER130, right panel: MamRep434). The deviation *z*-scores of TE accessibility in each cell were calculated based on the overlap between scATAC-seq peaks and the TE regions and mapped to the pseudo-time map obtained in Fig. 3. The colors indicate the deviation *z*-scores (red: high, purple: low). **c** Average TE accessibility scores along the putative differentiation axis. The pseudo-time of each branch ((S2, S1) and (S0, S1)) is equally divided into 10 segments, and the average TE accessibility scores were calculated for each segment. The *x*-axis shows the pseudo-time to the nodes of S2 (−0.027) and S0 (0.016), with branching point S1 at 0. The *y*-axis shows the average TE deviation *z*-score for each segment. **d** Two-group comparison of bias-corrected accessibility deviations of TE subfamilies between branches (S2, S1) and branches (S0, S1). The *x*-axis shows the difference in mean deviation scores between branch (S2, S1) and branch (S0, S1), and the *y*-axis shows −$\log_{10}(q - \text{value})$ using the Mann–Whitney *U* test with Benjamini–Hochberg correction. The values for each TE subfamily are shown in Supplementary Data. Black circles indicate each TE subfamily and red triangles indicate MER130 and MamRep434.

for the putative neuronal progenitor regulation, we compared the accessibility between pseudo-time branches for each TE subfamily. Specifically, we calculated the accessibility scores of each TE subfamily in each cell based on the overlap with the peaks of scATAC-seq for each TE subfamily instead of the k-mer in chromVAR[34]. Then we created a TE-by-cell accessibility matrix (see "Methods"). To search for TE subfamilies that were specifically accessible in branch (S2, S1), which was estimated as the state of neuronal progenitor cells (Fig. 3, Supplementary Fig. 7), we compared each of the TE accessibility scores between branch (S2, S1) and branch (S1, S0) (Fig. 5d). Thus, we found that among the TE subfamilies enriched for Neurod2 binding sites ($ES_{peak} > 1.5$ and $ES_{motif} > 1.5$), MER130, MamRep137 (DNA class in Repbase[51]), and MER102b (DNA class in Repbase[51]) are significantly more accessible in branch (S2, S1) (*q* < 0.05, FDR) (Fig. 5a, red triangles in the left panel). Similarly, among the TE subfamilies enriched for Lhx2 binding sites ($ES_{peak} > 1.5$ and $ES_{motif} > 1.5$), MamRep434 and LFSINE_Vert (SINE class in Repbase[51]) are significantly more accessible in branch (S2, S1) (*q* < 0.05, FDR) (Fig. 5a, red triangles in the right panel). We also investigated the number of TE-derived accessible motifs for each TE subfamily. Among the ATAC peaks with TE-derived de novo motif 1, 22/4744 were from MER130, 8/4744 were from MamRep137, and 4/4744 were from MER102b. Among the ATAC peaks with TE-derived de novo motif 3, 8/930 were from MamRep434, and 3/930 were from LFSINE_Vert. The details are shown in Supplementary Data.

To identify TE subfamilies that can directly contribute to the acquisition of binding motifs, we performed motif detection in consensus TE sequences using FIMO[53]. As a result, de novo motif 1 (Neurod2-like motif) and de novo motif 3 (Lhx2-like motif) were detected in the consensus TE sequences for MER130 and MamRep434, respectively (Fig. 6a–c). Notably, our result for MER130 is consistent with that of a previous report showing that MER130 has a putative consensus sequence for Neurod/Neurog binding and functions as enhancers during mouse neocortical development[26]. To ask whether de novo motif 1 in MER130 is the same as the previously determined Neurod/g consensus motif in MER130, we further characterized the de novo motif 1 sequence in MER130. We confirmed that the relative position of de novo motif 1 in the consensus sequence of MER130 was 216–222, which was the same as the position of the Neurod/g motif determined previously (Fig. 6b, c highlighted with orange bar). Then, to confirm whether the MER130-derived de novo motif 1 accessible in the mouse genome corresponds to the Neurod/g motif in the consensus sequence of MER130, the accessible motifs detected in the MER130 fragments in the genome were converted into the relative position in the MER130 consensus sequence (see "Methods"). The result showed that the accessible motifs were mainly mapped to the putative Neurod/g motif in the MER130 consensus sequence (Fig. 6d). Moreover, all MER130 fragments that harbor accessible de novo motif 1 contained the putative

Neurod/g motif of the MER130 consensus sequence. This result suggests that the motif detection site in the consensus MER130 sequence basically corresponds to the accessible motif sites in the genomic MER130 fragments and that some MER130 have additional motifs. We further investigated the surrounding sequence of de novo motif 1 in the genomic MER130 fragments. We performed multiple alignments of MER130 fragments, which overlap with the ATAC peaks and de novo motif 1 derived from the consensus TE sequence (*n* = 22, see "Methods"). The result showed that de novo motif 1 was selectively conserved among the accessible MER130 fragments when compared with the flanking regions within MER130 (Fig. 6a). Notably, we also observed another highly conserved tandem motif (GGCA and GCCA), which was annotated as a putative Nfi dimer motif in the previous study[26], at an upstream region of de novo motif 1 (Fig. 6a). Moreover, we aligned all of the accessible MER130 fragments with or without de novo motif 1 (*n* = 26) and confirmed that these motif sites tended to be conserved among these regions (Supplementary Fig. 9). These results suggest that de novo motif 1 in MER130 is the same as the previously determined Neurod/g consensus motif in MER130. Similarly, we also performed these experiments to characterize de novo motif 3 in MamRep434. We found that de novo motif 3 tended to be conserved among accessible MamRep434 fragments (Fig. 6a, Supplementary Fig. 9) and confirmed that accessible motifs in the genomic MamRep434 fragments were mainly mapped to the de novo motif 3 (Lhx2-like motif) in the MamRep434 consensus sequence (Fig. 6d, right panel).

From the results described above, we found that MER130 and MamRep434 had the highest enrichment scores for de novo motifs 1 and 3, respectively, and were significantly regulated in the putative neuronal progenitors (Fig. 5a, d). In addition, the de novo motifs were observed in their consensus sequences (Fig. 6). These results prompted us to investigate these two TEs in detail for further analysis. To ask to what extent MER130 and MamRep434 contributed to the accessible de novo motifs 1 and 3, respectively, we examined whether these TEs were enriched with the accessible de novo motifs compared to the whole TE population. We calculated the number of TEs that overlapped or did not overlap with the accessible de novo motifs and compared them between each TE (MER130 or MamRep434) and all TEs. The results showed that MER130 and MamRep434 were significantly more enriched with the accessible de novo motif 1 and 3, respectively, than all TEs (*q* < 0.01: FDR; Table 2). Then, to confirm that MER130 and MamRep434 were regulated in the putative neuronal progenitors, we visualized the accessibility of MER130 and MamRep434 on the putative differentiation trajectory map and calculated the average of the accessibility scores along the pseudo-time branches (S2, S1) and (S0, S1) (Fig. 5b, c). In both results of MER130 and MamRep434, the most substantial peaks of the accessibility scores were observed in the (S2, S1) branch that was characterized as a putative neuronal

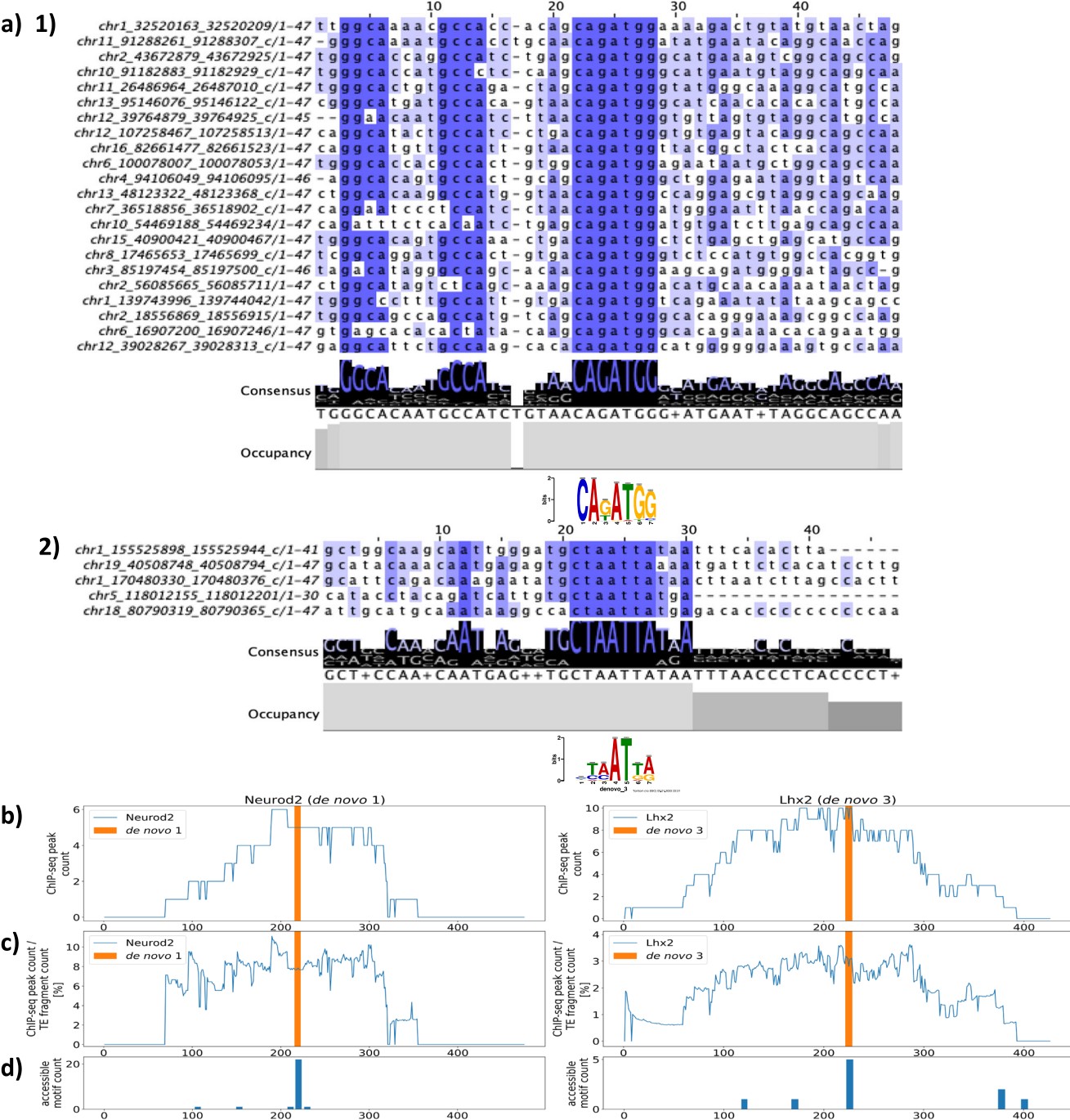

progenitor cell population. Together, these results suggest that MER130 and MamRep434 significantly contribute to a part of the cell-type-specific cis-elements and may have been acquired as transcription factor-binding sites that contribute to the function of glutamatergic neuronal progenitors.

**Specific TEs, including MER130 and MamRep434, can function as important cis-elements of neuronal development genes.** Next, we examined whether MER130 and MamRep434 could function as transcription factor-binding sites that are important for neuronal development. Since de novo motifs 1 and 3 were predicted to bind to Neurod2 and Lhx2, respectively (Fig. 2c), we mapped the ChIP-seq peaks of transcription factors in the genome to the consensus sequences of TEs based on RepeatMasker[54]

(see "Methods"). We used ChIP-seq data of Neurod2 and Lhx2 in the neural cell type after P0 in the ChIP-Atlas database[49]. The results showed that the ChIP-seq peaks were concentrated in the regions of approximately 200 bp around the de novo motifs detected in the consensus sequences (Fig. 6b, c). To account for the effect of input bias of ChIP-seq, we also evaluated the IP/input ratio for the proportion of reads that mapped to the detection sites of the TE-derived de novo motifs (see "Methods"). The IP/input ratios for de novo motif 1 in MER130 and de novo motif 3 in MamRep434 exceeded over 2-fold, suggesting that the transcription factor bindings were enriched at the detection sites of the motifs (Supplementary Fig. 10). These results suggest that the de novo motifs in these TEs play important roles as transcription factor-binding sites.

**Fig. 6 De novo motifs in MER130 and MamRep434 are targets of Neurod2 and Lhx2 binding, respectively. a** Multiple alignments of MER130 and MamRep434 fragments harboring de novo motifs accessible in the genome. The genomic instances of MER130 (panel (1), $n = 22$) or MamRep434 (panel (2), $n = 5$), which overlap with the pseudo-bulk ATAC peaks and de novo motif 1 or 3 derived from the consensus TE sequence, respectively, were aligned. Each panel shows the ± 20 bp of the consensus de novo motif in genomic TE fragments. The left column on each panel indicates the genomic position of each TE fragment, and the numbers above the alignments indicate arbitrary aligned positions. The shade of the blue highlight shows the degree of the matched nucleotide among aligned nucleotides in each position. The consensus annotation shows the representative nucleotide based on the proportion of aligned nucleotides in each position. The consensus sequence of the multiple alignments is indicated below the consensus annotation. The occupancy annotation shows the number of nucleotides aligned at each position in the multiple sequence alignments. The bottom motif logo shows the corresponding position of de novo motifs 1 and 3, respectively. **b, c** Distribution of transcription factor ChIP-seq peaks (left: Neurod2, right: Lhx2) mapped from the genome to consensus TE sequences. **b** The y-axis shows the number of peaks mapped to the consensus MER130 and MamRep434 sequences, respectively (blue lines). **c** The y-axis shows the percentage of the peaks normalized by the total number of MER130 and MamRep434 fragments in the genome calculated at 1 bp resolution, respectively (blue lines). In each panel, the x-axis indicates the position in the consensus TE sequence obtained from RepeatMasker (.align output). The de novo motifs detected in the consensus TE sequence are shown as orange highlights (de novo motif 1 was 216–222 bp in MER130 and de novo motif 3 was 222–228 bp in MamRep434). **d** Distribution of the detection sites of accessible de novo motifs mapped from the genome to consensus TE sequences. The y-axis shows the number of de novo motif detection sites in TE-derived ATAC peaks mapped to the consensus MER130 and MamRep434 sequences. The total number of the input genomic TE fragments used for the calculation was 22 (MER130) and 8 (MamRep434). Some of the TE fragments contained multiple de novo motifs. The x-axis shows the relative positions of the consensus sequence of TEs obtained from RepeatMasker (.align output).

**Table 2 MER130 and MamRep434 fragments tended to overlap significantly with accessible de novo motifs 1 and 3, respectively.**

| Motif | Condition | Overlap with accessible motifs | Not overlap with accessible motifs |
|---|---|---|---|
| de novo 1 | MER130 | 22 | 68 |
| | All TEs | 4821 | 3,579,641 |
| de novo 3 | MamRep434 | 8 | 519 |
| | All TEs | 938 | 3,583,524 |

Each row indicates the number of MER130 or MamRep434 fragments and all TE fragments. Each column indicates the number of TEs that overlapped or did not overlap with accessible motifs. The cell for overlap with accessible motifs indicates the number of TE fragments in which the motifs detected inside the TEs were also included in the ATAC peaks.

We then investigated the function of genes located near the TEs. We obtained the genes near TE fragments in the genome using GREAT[55] and performed Gene Ontology analysis using g:Profiler[56]. As a result, the terms related to synaptic membrane, development, and morphogenesis were enriched in the genes near MER130 (Fig. 7a, b). In addition, the terms related to neurogenesis were also enriched in genes near MamRep434 (Fig. 7a, c). Notably, whereas GO terms related to the synaptic membrane were common between MER130 and MamRep434, GO terms related to the higher structures of the brain, such as "anatomical structure development" and "axon development", were only observed in the GO terms for the MamRep434 neighboring genes (Fig. 7a–c). These results suggest that MER130 and MamRep434 themselves might be inserted into the gene neighborhoods related to neuronal differentiation during evolution. In addition, to identify the genes targeted by TE-derived cis-elements, we focused on the genes next to the TE-derived accessible de novo motifs in the genome using GREAT[55]. As a result, MER130-derived accessible de novo 1 motifs were located next to the genes whose perturbations cause abnormalities in telencephalic morphology (*Ap3b1*, *Dgkb*, *Id4*, and *Prok2*; previously reported in ref. [26]), and the *Kcnj3* gene, which has GO terms associated with components of the synaptic membrane. MamRep434-derived accessible de novo 3 motifs were located next to the genes *Btbd3*, *Nos1*, and *Pbx1*, which have GO terms associated with neurogenesis. These genes have been reported to have neuronal developmental functions related to the regulation of cortical layer structures developed in mammals[57–60]. Thus, in

the vicinity of the TE-derived accessible motifs, there were candidate target genes that may be related to the functional evolution of the brain. The other candidate target genes located near TE-derived cis-elements are listed in Supplementary Tables 3 and 4.

We also checked whether other TEs identified as the candidates for transcription factor binding in the putative neuronal progenitors could function as cis-elements (that is, MamRep137 and MER102b for Neurod2 binding, and LFSINE_Vert for Lhx2 binding) (Fig. 5a red triangles). We converted the ChIP-seq data of Neurod2 or Lhx2 to the TE consensus sequences and found that the ChIP-seq peaks were enriched in each TE consensus sequence (Supplementary Fig. 11a(1), a(2), b(1), b(2)). This result suggests that these transcription factors can bind to each TE subfamily. Then, to compare the ChIP-seq data and the relative position of the accessible de novo motifs (transcription factor-binding motifs), we mapped the accessible motifs in the genomic TEs to the TE consensus sequences (Supplementary Fig. 11a(3), b(3)). We found that the accessible motifs of the genomic MamRep137 or LFSINE_Vert were mainly mapped to one position near the center of the ChIP-seq peak in the consensus sequence. In contrast, accessible motifs of the genomic MER102b were mapped to multiple sites around the ChIP-seq peak. These results suggest that these TEs also could contribute to the transcription factor-binding sites through the de novo motifs.

**MER130 and MamRep434-derived cis-elements may contribute to brain evolution.** Finally, we estimated the acquisition time of cis-elements derived from MER130 and MamRep434 during evolution. To estimate the time of TE-derived cis-element acquisition, we searched for the presence/absence of orthologous sequences of ATAC peaks that overlapped with TEs (see "Methods"). Figure 7d showed that the TE-derived cis-elements were mainly acquired at two stages: the ancestor of Amniota (105–159 Mya) and Muridae (29–73 Mya), which is consistent with the results of a previous report[28].

For individual TE subfamilies, our results suggest that MER130- and MamRep434-derived cis-elements were acquired at different times (Fig. 7d). Our results showed that the main acquisition of MER130-derived cis-elements occurred in the mammalian ancestor, and 42% of the ortholog sequences of MER130-derived cis-elements were detected in the ancestor of Amniota (105–159 Mya). In contrast, the main acquisition of MamRep434-derived cis-elements occurred in the Eutherian ancestor, and 67% of the MamRep434-derived cis-elements were

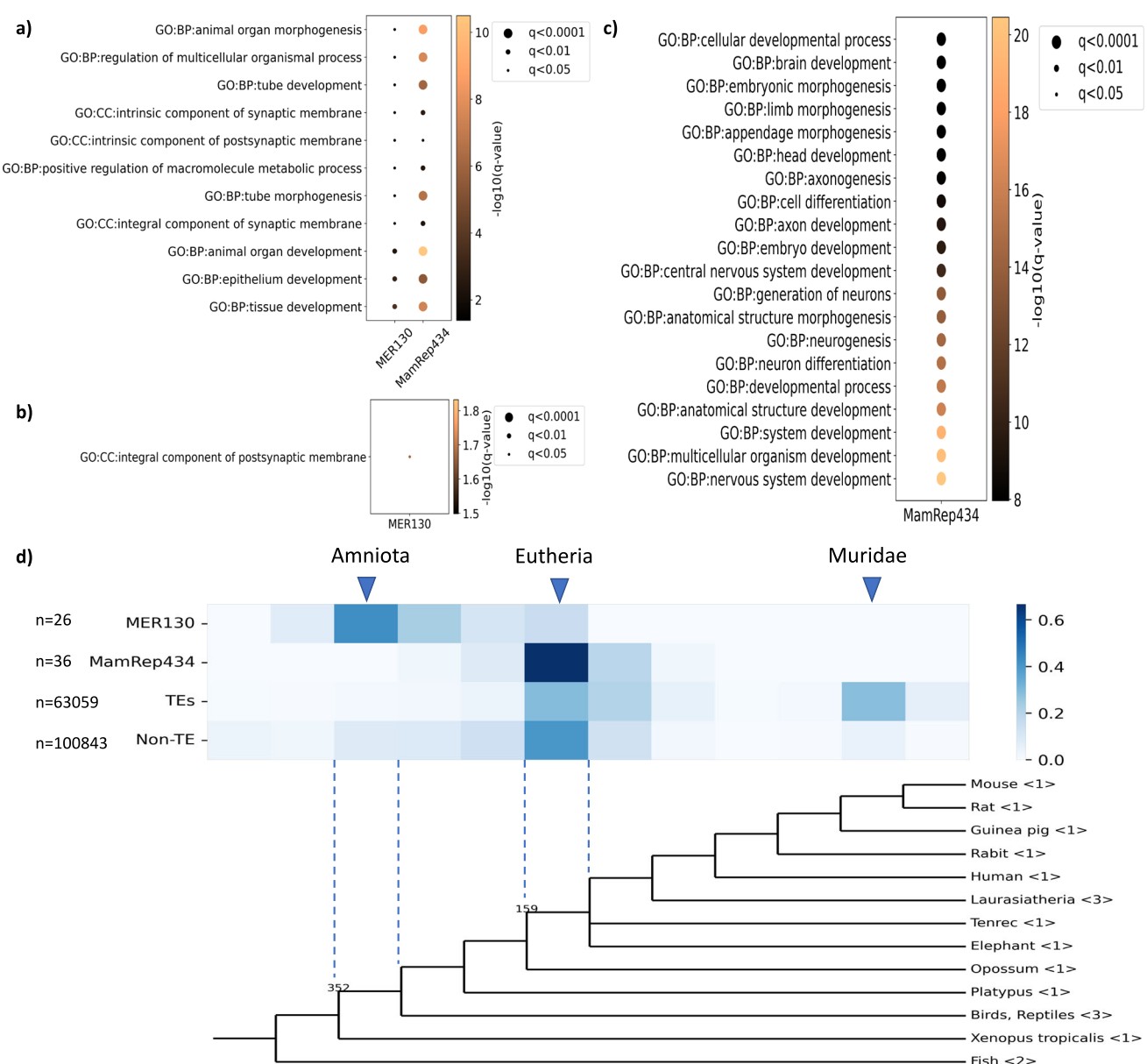

**Fig. 7 MER130 and MamRep434-derived cis-elements are associated with the function of transcription factors and acquired at different times.**
**a–c** Gene ontology (GO) terms enriched in TE-neighboring genes (FDR, q < 0.05). **a** GO terms commonly enriched in MER130 and MamRep434 neighboring genes and included in the top 20 are shown. **b**, **c** Up to 20 terms enriched only in **b** MER130 or **c** MamRep434 neighboring genes are shown. The color depth of the plots indicates the enrichment score in $-\log_{10}(q-\text{value})$, and the size was determined according to (q-value). **d** The estimated time of acquisition of TE-derived cis-elements during evolution. The heatmap shows the percentage of peaks assigned to that clade out of the total number of bulk ATAC peaks (n; indicated on the left side). Bulk ATAC peaks were obtained in "Data sources''. The number of species used for analysis in each clade is indicated by ⟨⟩. The values on the branches are divergence times (in a million years ago, Mya).

acquired in the ancestor of Eutheria (312–352 Mya). We performed Fisher's exact test to ask whether the acquisition time of these TE-derived cis-elements was independent. A comparison of the number of TE-derived cis-elements acquired in the ancestors of Amniota and Eutheria showed that they significantly differed between MER130 and MamRep434 (p < 0.01, Fisher's exact test). These results were also similar for each cell type and differentiation process of the excitatory neurons (Supplementary Fig. 12), suggesting that MER130- and MamRep434-derived cis-elements were acquired mainly in the ancestors of Amniota and Eutheria, respectively. These may reflect differences in the timing of acquisition of the TEs themselves, with Amniota as the ancestor of MER130 and Mammalia as the ancestor of

MamRep434 in Repbase[51]. Our results showed the most distantly related species in which 7.7% of the ortholog sequences of MER130-derived cis-elements were detected was *Xenopus tropicalis* (no orthologous sequences were detected in fish), which is consistent with the results of Notwell et al.[26]. In addition, the most distantly related species in which 2.7% of the ortholog sequences of MamRep434-derived cis-elements were detected was platypus, in the same time as the origin considered in Repbase[51]. To determine whether the de novo motifs play an evolutionarily important role as functional domains in the MER130- and MamRep434-derived cis-elements, we also evaluated the sequence conservation of the motifs in placental mammals (see "Methods"). Among the MER130- and MamRep434-derived

cis-elements in the mouse genome, motif sequences were highly conserved in mammals (Supplementary Fig. 13). This suggests that the motifs are conserved as functional domains in TE-derived cis-elements.

## Discussion

Accumulating evidence from recent papers suggests that TEs are one of the important sources of enhancers and contribute to the evolution of species, conferring new regulatory networks of transcription factors[21,25,61]. Based on this, we hypothesized that during evolution, some TEs contribute to having specific cell populations or regulating the function of particular populations in the mouse brain (Fig. 1). Thus, in this study, we aimed to find important TEs that contribute to cell-type-specific regulation in the mouse adult brain. To this end, we first investigated the most variably accessible motifs of cis-elements across cell types, which could characterize specific cell populations at single-cell levels using scATAC-seq data. Because we considered that some TE-derived cis-elements could be regulated in unknown populations that are not characterized enough, we preferred an unbiased method that does not depend on known cell types. Therefore, we used scATAC-seq data to characterize unbiased cell populations and their regulatory cis-elements. Accordingly, we analyzed the most variably accessible motifs, which were regulated in a cell-type or cell-population-dependent manner using scATAC-seq data of the mouse adult brain with chromVAR[34] (Fig. 2). Consequently, we obtained the most variable four de novo motifs as candidates. Interestingly, pseudo-time analysis showed that de novo motif 1 (Neurod2-like motif) and de novo motif 3 (Lhx2-like motif) were accessible in the branch of putative glutamatergic neuronal progenitor cells, especially those suggested to be related to stress-responsive neurogenesis for tissue recovery (Fig. 3). In fact, Neurod2 and Lhx2 have been reported to function in the neurogenesis of glutamatergic neurons[46–48]. In addition, Neurod2 has been proposed to function as a nexus for stress-related dendritic synaptic plasticity[46]. This suggests that de novo motif 1 and de novo motif 3 were regulated in a putative neuronal progenitor cell population, which was not labeled in the original scATAC-seq data. Although this population was still not characterized well in the previous report[33], we speculate that this population consisted of neuroblasts migrating from SVZ or other neuronal progenitors with unknown origins[62–64]. Subsequently, we focused on these motifs and found that two TEs, MER130 and Mam-Rep434, were enriched in de novo motif 1 and de novo motif 3, respectively. In addition, they were enriched in transcription factor-binding sites of Neurod2 and Lhx2, respectively (Fig. 5a). The results of ChIP-seq peaks mapping to consensus sequences of the TEs suggest that MER130 and MamRep434 induce Neurod2 and Lhx2 binding based on their internal motifs (Fig. 6). In addition, MER130 and MamRep434 TEs themselves were significantly accessible in a putative neuronal progenitor cell population (Fig. 5b–d), suggesting that MER130- and MamRep434-derived cis-elements serve as open chromatin for the binding of Neurod2 and Lhx2 transcription factors, especially in neuronal progenitors in the mouse adult brain. Thus, our approach has enabled us to find unique TEs that contribute to cell-type-specific regulation, especially in the context of adult neurogenesis.

Interestingly, a previous report has shown that MER130-derived cis-elements can function as enhancers for NPCs during embryonic neocortical neurogenesis, harboring putative transcription factor-binding sites essential for neuronal differentiation, including Neurod/g[26]. In this study, we found that the MER130-derived cis-elements were also active in adult neurogenesis. Indeed, comparing our result of MER130 fragments

overlapping ATAC peaks with their results, we were able to cover most of the MER130 fragments they detected with our results (18/23 fragments), and we detected some additional MER130 fragments (18 fragments) with the exact condition of the TE annotation as that found in the previous study (Supplementary Fig. 14). This suggests that many of the MER130-derived cis-elements were commonly accessible both in embryonic and adult neurogenesis, and they might play a fundamental role in neurogenesis. The difference between the data may reflect the different methods for detecting active sites (p300 binding vs scATAC-seq signal) and/or distinct mechanisms between embryonic and adult neurogenesis. Further studies are required to investigate the difference in MER130 function between embryonic and adult neurogenesis. Furthermore, it would be interesting to investigate the differences between other TEs that contribute to the neurogenesis of embryonic and adult mouse brains. Moreover, comparisons between the brains of multiple species, including mice and humans, would be essential to understand the roles of TEs from the aspect of evolution. These themes will be the next question of this study.

Importantly, we found that the transcription factor-binding motifs (that is, Neurod2 motif in MER130 and the Lhx2 motif in MamRep434) tended to be conserved not only among the TE subfamily accessible in the mouse genome but also among that of mammalian species (Fig. 6a, Supplementary Fig. 9, Supplementary Fig. 13). Given that these motifs are also observed in the TE consensus sequences, these TEs could originally have had consensus motifs for transcription factors, and the motifs could have been protected from gaining mutations during evolution through natural selection. In contrast, we could not detect transcription factor motifs in the TE consensus sequences of MamRep137 and MER102b, which were enriched for the Neurod2 transcription factor, and LFSINE_Vert which was enriched for the Lhx2 transcription factor (Supplementary Fig. 11). These TEs might acquire transcription factor-binding sites by accumulating mutations in pre-motifs in each site during evolution. This notion is consistent with that in a recent paper suggesting that independent TEs can evolve into enhancers with the same function through convergent evolution, mainly observed in redundant enhancers[65]. This finding indicates that the transcriptional network is formed not only in the TE groups that are considered to have originally had motifs but also in other TE groups that are considered to have later acquired motifs during evolution.

According to the functional analysis of the TE-neighboring genes in the mouse genome (Fig. 7a–c) and the acquisition time of TE-derived cis-elements (Fig. 7d), MER130 may be acquired in pre-mammalian species to perform fundamental cortical functions such as synaptic membrane regulation or morphogenesis, which is consistent with previous studies[26,27]. In addition, we suggest that MamRep434 may be responsible for neurogenesis-related genes, which are related to the regulation of cortical layer structures developed in Eutheria. In fact, in Eutheria, the production of neurons is more active than in other mammals, such as Metatheria[66]. The high conservation of motifs in the TE-derived cis-elements suggests that TE insertion contributes to the acquisition of motifs as functional domains. Furthermore, there is an association between the function of transcription factors whose binding is enriched in TEs and the function of TE-neighboring genes. The Neurod2 transcription factor, which is enriched in MER130, has been reported to be involved in the regulation of synaptic innervation and spine morphogenesis[46,67,68]. The Lhx2 transcription factor, enriched in MamRep434, has been reported to have several roles in the spatiotemporal regulation of neurogenesis and is considered a key factor for brain structure and neuronal identity[8]. For example, Lhx2 plays a role in the spatiotemporal regulation of neuron number by promoting a

decrease in the number of neurons in layer V and an increase in the number of neurons in layer VI[69]. Interestingly, the timing of acquiring TE-derived cis-elements is associated with the function of transcription factors, which might relate to the evolutionary changes in the brain structure. This is consistent with our hypothetical scenario that TEs confer new cis-elements of transcription factors for specific cell types in multiple stages and contribute to gaining higher complexity in the brain during evolution. Our results suggest that MER130- and MamRep434-derived cis-elements have distinct functions and are acquired at different times during evolution. This may reflect the gradual evolution of the mammalian brain architecture and the acquisition of complex functions in the brain by utilizing TE-derived regulatory elements of transcription factors related to neuronal development.

## Conclusion

MER130 TEs, acquired by the ancestor of Amniota, serve as binding sites for the transcription factor Neurod2 and promote glutamatergic neuronal progenitor cell-specific gene regulation. MamRep434 TEs, acquired by the ancestor of Eutheria, also serve as binding sites for the Lhx2 transcription factor and promote glutamatergic neuronal progenitor cell-specific gene regulation.

## Methods

**Data sources**. To obtain accessible regions for transcription factor binding per cell, we used scATAC-seq of the adult mouse (P56) prefrontal cortex, which is alternatively available via accession number: GSE111586[33]. In this study, among the cell-type labels defined by Cusanovich et al.[33], we used scATAC-seq data of the labels including more than 100 cells as follows: callosal projection excitatory neurons (Ex._neurons_CPN: layer II/III excitatory neurons), subcerebral projection excitatory neurons (Ex._neurons_SCPN: layer V excitatory neurons), corticothalamic projection excitatory neurons (Ex._neurons_CThPN: layer VI excitatory neurons), SOM+_Interneurons (somatostatin positive interneurons), Inhibitory_neurons, Oligodendrocytes, Microglia, and Astrocytes. We first split the scATAC-seq data into bam files for each cell-type label and performed peak calling using MACS2[70] with the following parameters: "- nonmodel-keep-dup all-extsize 200-shift -100" for each cell population. To create pseudo-bulk ATAC-seq peak data, we merged the peak sets for each cell population and resized the peaks to a width of 500 bp. In addition, we obtained a peak-by-cell chromatin accessibility matrix based on the fragment counts for each cell in the region corresponding to the bulk ATAC peaks.

Using scATAC-seq data, we also obtained gene activity scores calculated based on chromatin accessibility near each gene provided by Cusanovich et al.[33]. Gene activity scores were normalized for individual cells according to size factors.

To obtain gene expression levels in the cell types corresponding to scATAC-seq data, we used scRNA-seq data of the adult mouse cortex downloaded from https://www.dropbox.com/s/cuowvm4vrf65pvq/allen_cortex.rds?dl=1, which is alternatively available via accession number: GSE71585[13].

We also used ChIP-seq peak data of neural cell-type classes from ChIP-Atlas[49] to identify transcription factor-binding sites and histone modification sites (the threshold for significance of peak calling with MACS2[70] was set to $-10\log_{10}$ [MACS2 Q-value] = 50). Of the data available in the ChIP-Atlas, we used only the neural cell-type class after P0. The SRA accession numbers of the sequencing data used in this study are shown in Supplementary Table 5.

In addition, to obtain known transcription factor-binding motifs, we used the motifs provided by the HOCOMOCO v11 database[71].

For the TE annotations, we used the mapping result of Repeat Library 20090604 to mm9, which was published by RepeatMasker[54]. In this study, simple repeats, satellite DNAs, and small RNAs were excluded from the TE annotations.

**Calculation of k-mer/motif accessibility**. We used the computeDeviations function of chromVAR[34] to obtain the bias-corrected accessibility deviations as k-mer/motif accessibility profiles based on the presence or absence of the k-mer/motif in each ATAC peak. In this analysis, k-mer/motif annotation for each peak and peak-by-cell accessibility matrix (described in "Data sources") were inputted to computeDeviations function. For k-mer annotation, we used the 7-mer annotation. In eukaryotes, the most common core motif to which transcription factors bind is 7 bp in length[72]; therefore, we chose the 7-mer in this study, which is also the default length for using chromVAR[34] in the STREAM paper[42]. For motif annotation, motifs were detected in the ATAC peaks with the matchMotifs function (default parameter) in the motifmatchr package as part of chromVAR[34]. In this study, since the analysis was performed at the single-cell level, we used ATAC-seq as epigenomics data.

To visualize the similarity among cells for k-mer accessibility profiles in Fig. 2a, t-SNE was performed using the deviationsTsne function (default parameter) of chromVAR[34]. In Figs. 2b, 3c, d and Supplementary Fig. 7a, the deviation z-scores of each k-mer/motif were also calculated with the computeDeviations function (default parameter) of the chromVAR[34].

**De novo motif generation based on k-mer accessibility**. Using chromVAR's assembleKmers function (the threshold of variability score = 2.0)[34], we generated de novo motifs by weighting the k-mers with similar accessibility patterns to the seed k-mer with large accessibility variation across the cells. The k-mer accessibility profiles (k = 7) calculated with the computeDeviations function described in the "Calculation of k-mer/motif accessibility" were used as inputs.

**Motif comparison**. We calculated the similarity of the obtained de novo motifs to known motifs in Fig. 2c and Supplementary Fig. 3. Using TOMTOM (with Pearson correlation coefficient as the motif column comparison function)[35], which compares motifs based on the frequency of occurrence of letters at each position in the sequence, we searched the HOCOMOCO v11 database[71] for known motifs that were significantly similar to the de novo motifs.

**Assignment of scATAC-seq cell-type labels to scRNA-seq data**. To estimate the cells of scRNA-seq data corresponding to the cell-type labels of scATAC-seq data, we transferred cell-type labels with Seurat[73]. We used the FindVariableFeatures function (default setting) to select genes with considerable activity variation across the cells based on scATAC-seq. Then, using the FindTransferAnchors function (default setting), the cell-type labels defined in scATAC-seq were assigned to each cell in scRNA-seq based on the similarity of gene activity/expression patterns for the selected genes.

**Pseudo-time analysis based on k-mer accessibility**. To reconstruct the differentiation process of cells labeled as neurons by Cusanovich et al.[33], we used STREAM[42] for pseudo-ordering the cells based on the k-mer accessibility profiles. For pseudo-time analysis, we followed the original pipeline, and each cell was mapped to a two-dimensional tree structure called a flat tree plot calculated using STREAM in Fig. 3.

**Comparison of TE-ATAC peak overlap between cell types using randomly sampled cells**. To investigate the overlap with TE for each cell type, we compared the overlap with TE using pseudo-bulk ATAC peaks obtained from randomly sampled cells of each cell type in Supplementary Table 1. First, 197 cells were randomly obtained from each cell type, aligned to the microglia with the lowest number of cells from scATAC-seq data. Then, peak calling was performed using MACS2[70] with the following parameter:"-nomodel-keep-dup all-extsize 200-shift -100" as in "Data sources", and the length of each peak was aligned to 500 bp to create bulk ATAC peaks for each cell type.

**Enrichment of TE subfamilies in transcription factor ChIP peaks**. To identify TE subfamilies that were enriched in transcription factor-binding sites, we calculated the enrichment of ChIP peaks for TEs in the genome in Fig. 5a. Specifically, we defined the enrichment of transcription factor binding to TEs as

$$ES_{peak} = \log_2\left(\frac{\frac{\text{Length of overlap between TF peaks and TE subfamily}_{(bp)}}{\text{Length of TF peaks}_{(bp)}}}{\frac{\text{Length of TE subfamily}_{(bp)}}{\text{Genome Length}_{(bp)}}}\right). \quad (1)$$

The intersectBed tool of the BEDTools package[74] was used to obtain the overlap regions between TF ChIP peaks and the TE subfamily. The enrichment score quantitatively evaluated the degree of overlap between the TE and ChIP peak; TEs with less overlap with ChIP peaks have a relatively low score. In cases where multiple TEs overlap with the same ChIP peak, the overlap between each TE and the peak was separately calculated in base pair (bp) units using the intersectBed tool. To prevent the enrichment score from being abnormally high due to a small number of TE fragments, we restricted the subfamilies to TEs that intersected the ChIP peak at least three times in the whole genome. To calculate statistical significance, Fisher's exact test with the Benjamini–Hochberg correction was performed based on the numbers used to calculate the enrichment scores.

**Detection of TE-derived accessible motifs**. The motifs in the ATAC peaks were detected using chromVAR's matchMotifs function (default parameter)[34], and those motifs that were also included in TEs were identified as TE-derived accessible motifs. In this analysis, we used bulk ATAC-seq data obtained by merging the scATAC-seq peaks (described in "Data sources").

**Enrichment of TE subfamilies in accessible motifs**. We also calculated the enrichment of accessible motifs for TEs in the genome in Fig. 5a. Specifically, we

defined the enrichment of transcription factors binding to TEs as

$$ES_{motif} = \log_2 \left( \frac{\dfrac{\text{Number of ATAC peaks with motif in TE subfamily}}{\text{Number of ATAC peaks with motif}}}{\dfrac{\text{Length of TE subfamily}_{(bp)}}{\text{Genome Length}_{(bp)}}} \right). \quad (2)$$

The intersectBed tool in the BEDTools package[74] was used to obtain the ATAC peaks where the entire motif was detected. Subsequently, as "number of ATAC peaks with motifs in the TE subfamily" in (2), we counted the number of ATAC peaks containing the entire motif, which is included in both the TE and the ATAC peaks. To prevent the enrichment score from being abnormally high due to a small number of TE fragments, we restricted the subfamilies where ATAC peaks and motif-containing TEs intersected at least three times in the whole genome. To calculate statistical significance, Fisher's exact test with the Benjamini–Hochberg correction was performed based on the numbers used to calculate the enrichment scores. To account for technical biases of scATAC-seq (GC bias, average accessibility and fraction of reads in peaks), the TE enrichment score was corrected for the background peak set calculated with the getBackgroundPeaks function (default parameter) of chromVAR[34]. The accessible motifs were assigned to the positions within the background peak calculated with chromVAR while maintaining the relative position of the motifs in the peaks. We calculated the enrichment score for each background peak set (as a control) and used them to compute the robust z-score.

**Calculation of TE accessibility**. Using the TE subfamily as annotations instead of the k-mer/motif, we calculated the TE subfamily accessibility scores for each cell with chromVAR[34]. We used the getAnnotations function of chromVAR to obtain annotations where ATAC peaks intersect TE regions by more than 1 bp in the genome. Then, the annotation and peak-by-cell accessibility matrix (described in "Data sources") were input to compute the deviation function as in "Calculation of k-mer/motif accessibility" to calculate the deviation scores of the TE subfamily. In this analysis, to exclude cases in which TEs were assessed as abnormally accessible, we restricted our analysis to the TE subfamily, in which more than 100 neuron-labeled cells overlapped with at least one ATAC peak. All neuron-labeled cells in the scATAC-seq data were used for the calculation. The Mann–Whitney U test with Benjamini–Hochberg correction was performed to test for statistical significance.

**Multiple alignments of TE fragments**. Multiple alignments were performed for TE fragments overlapping ATAC peaks in the genome. TE fragments of MER130 or MamRep434 overlapping with the pseudo-bulk ATAC peaks by more than 1 bp were collected from the alignment results between the mouse genome (mm9) and the consensus TE sequence obtained from RepeatMasker (.align output). In Fig. 6a, the TE fragments with the consensus de novo motifs were aligned to the consensus TE sequence and displayed ±20 bp of the consensus motifs. In Supplementary Fig. 9, multiple alignments were performed for TE fragments with or without the consensus de novo motifs. Multiple alignments were performed on these TE fragments using Mafft (default setting)[75] and visualized with Jalview[76]. The genomic regions where TEs were detected as the anti-sense strands were converted to the sense sequences to align them with other sequences.

**Mapping of ChIP-seq peaks to consensus TE sequences**. To detect binding target sites of transcription factors in TE sequences, we mapped ChIP-seq peaks in the genome to consensus TE sequences. The ChIP-Atlas database provides processed ChIP-seq peaks of multiple experiments[49]. In this experiment, we converted ChIP-seq peaks mapped to the TEs in the genome into the consensus TE sequences. ChIP-seq data used in this experiment was indicated in the Data Sources section ("Data sources"). The regions in the consensus TE sequence corresponding to the ChIP peak regions (Fig. 6b) or the TE fragments (Fig. 6c) in the genome were counted based on alignment results between the mouse genome (mm9) and the consensus TE sequence obtained from RepeatMasker (.align output)[54].

To account for ChIP-seq inputs, we also evaluated the IP/input ratio for the proportion of reads mapped to the sites of TE-derived de novo motifs detected with chromVAR[34] in the genome. The plotenrichment function (default setting) of deeptools[77] was performed with the bam file of ChIP-seq reads mapped using bowtie2[78] as the input, and the ratio of featureReadCount / totalReadCount was compared for each IP and input file in Supplementary Fig. 10.

**Mapping of TE-derived accessible motif detection sites to consensus TE sequences**. To detect sites in the consensus TE sequence corresponding to TE-derived accessible motifs in the genome, we mapped the sites of TE-derived accessible motifs detected with chromVAR[34] to consensus TE sequences. In Fig. 6d and Supplementary Fig. 11, the regions in the consensus TE sequence corresponding to the TE-derived accessible motif detection sites in the genome were counted based on alignment results between the mouse genome (mm9) and the consensus TE sequence obtained from RepeatMasker (.align output)[54].

**Motif scanning in consensus TE sequences**. FIMO[53] was used to detect individual motifs in consensus TE sequences. In this study, de novo motifs generated with chromVAR[34] were used as inputs for motif detection in consensus TE sequences.

**Gene Ontology analysis of associated genes**. GREAT[55] was used to associate genomic regions with neighboring genes (default regulatory domain definition; basal domain that extends 5 kb upstream and 1 kb downstream from the transcription start site, and an extension up to the basal regulatory domain of the nearest upstream and downstream genes within 1 Mb). Gene Ontology analysis of gene lists was performed using g:Profiler[56]. g:Profiler outputs GO terms that are significantly enriched ($q < 0.05$) in the input genes list.

**Ortholog search for the cis-elements**. The orthologous sequences of the ATAC peak were searched using the liftOver tool using the cross-species chain data obtained from the UCSC Genome Browser database[79]. Using ATAC peaks in the mouse genome as input, we estimated the acquisition time of the cis-elements by checking for the presence of orthologous sequences in other species (minimum ratio of bases that must remap: 0.5). Among the species for which orthologs were identified, each ATAC peak was assigned to the species most distantly related to the mouse in the reference phylogenetic tree[28,80–83] and used to estimate the time of sequence acquisition. The following species were searched as orthologs: Rat (rn5), Guinea pig (cavPor3), Rabbit (oryCun2), Human (hg19), Cow (BosTau7), Dog(-CanFarm2), Horse (EquCab2), Tenrec (echTel1), Elephant (loxAfr3), Opossum (monDom5), Platypus (ornAna1), Chicken (galGal3), Turkey (melGal1), Lizard (anoCar2), X. tropicalis (xenTro3), Zebrafish (danRer7), Medaka (oryLat2). For the TE-derived cis-elements, the ATAC peaks that overlapped with the TE annotation of RepeatMasker were used as input in Fig. 7d.

**Sequence conservation in TE-derived cis-elements**. Sequence conservation scores (mm9.30way.phyloP30way.placentalMammals) in placental mammal genomes per bp were obtained from the UCSC Genome Browser database[79,84]. Average sequence conservation scores were obtained for each TE-derived accessible motif set. As a control, 7-bp sequences from the intersection of the ATAC peaks and the TE regions were randomly obtained 500 times with the shuffleBed tool (with the -noOverlapping option) of the BEDTools package[74]. At this time, only sequences that did not overlap with the motif detection sites were used as controls.

**Statistics and reproducibility**. The type of statistical tests used are described in the figure legends or the main text. To ensure reproducibility, we provided the source code on GitHub (described in the "Code availability" section).

**Reporting summary**. Further information on research design is available in the Nature Portfolio Reporting Summary linked to this article.

## Data availability

The accession numbers of the sequencing data used in this study are shown in Supplementary Table 5. Source data of the Figs. 4 and 5a, d are presented in Supplementary Data.

## Code availability

The code is available at https://github.com/hmdlab/TE_analysis.

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

## Acknowledgements

Computation for this study was performed in part on the NIG supercomputer at the ROIS National Institute of Genetics. We thank the members of the Hamada Laboratory for their valuable comments. This work was supported by JSPS KAKENHI [grant numbers 16H06279, 16H05879, and 20H00624 to M.H. and 21K06133 to M.O.] and AMED [grant numbers JP22ama121055, JP21ae0121049, and JP21gm0010008 to M.H.].

## Author contributions

H.M. and M.O. supervised the project. M.O. and K.S. contributed to the conception and experimental design. K.S. acquired and analyzed the data. H.M., M.O., and K.S. interpreted and discussed the results. M.O. and K.S. contributed to the manuscript drafting. All authors critically reviewed and revised the manuscript draft and approved the final version for submission.

## Competing interests

The authors declare no competing interests.
