## [Peer Review File · Communications Biology]

Reviewers' comments:

Reviewer #1 (Remarks to the Author):

In this work, Sekine et al. performed a comprehensive analysis of publicly-available scATAC-seq data from adult mouse cortex to explore cell-type specific cis-elements derived from transposable elements (TEs). The results clearly indicate that MER130 and MamRep434 are contributors that have provided the binding sites of Neurod2 and Lhx2, respectively, in neuronal progenitor cells. The study presents a nice correlation among accessible binding motifs, neuronal cell type, and TEs that contain the binding motifs. The manuscript is well-organized, and the figures and tables are of high quality and easily interpretable. The present report will undoubtedly appeal to wide readers of researchers and is well within the scope of Communications Biology. However, the manuscript lacks a description of the previous finding for putative NeuroD-binding sites derived from MER130. The authors need to address the following points.

1) The authors should note that a previous study, Notwell et al. (Nat. Comm. 2015, doi: 10.1038/ncomms7644), have also reported about MER130 that contain putative Neurod-binding sites to which p300, an enhancer co-activator, binds in the neocortex of mouse embryos. Although the previous paper has been cited in the manuscript, the authors did not mention that the MER130-derived putative Neurod-binding sequences have been reported previously. Particularly, Notwell et al. (2015) showed that the putative Neurod motif is present in around 230 bp of the MER130 sequence (see Fig. 2 of Notwell's paper), and I do not understand whether or not this is the same as the position of motif1 in 200-300 bp of MER130 reported in the manuscript (Figure 5c and 5d). The authors need to compare the results and add descriptions of the previous findings in the manuscript.

2) Page 5, line 134-135; "H3K4me1, which is known as an enhancer label"
A reference should be provided in this sentence.

3) Page 8, lines 183-184; "First, we mapped the ChIP-seq reads of transcription factors to the consensus sequences of TEs based on RepeatMasker"

Did the authors map the ChIP-seq reads directly to the consensus sequences of TEs? Or the ChIP-seq reads were mapped to the genome, and then the binding regions were converted to the TE consensus sequences? I think the latter is correct, but this sentence implies the former. Please clarify.

4) The authors should state that NDF2 of the HOCOMOCO motif logo in Fig. 2 is the same as Neurod2.

5) Legend to Figure 2; "De novo motifs"  "De novo motifs"

6) Font sizes of some figures are too small (e.g., Fig. 2, 3, and 5).

Reviewer #2 (Remarks to the Author):

Sekine and colleagues presented motif-analysis-based research to understand DNA context within the open chromatin regions(OCR) at the cell-type level. By reanalyzing the scATAC-seq of the mouse brain, the authors confirmed the concordance between OCRs and the K-mers signatures of OCR for the major cell types. They specifically found certain TFBS motifs enriched in different subfamilies of TE, and explored the potential evolution history of these insertion events. The presented results are interesting and solid. The approaches used in the study also reflect how the huge advantage and progress of single-cell experiment technology to facilitate computational data mining in answering biological questions. However, there are certain concerns to the current presented work.

Major concerns.

1. The central approach of this work relies on de novo motif finding. For this step, the authors simply adopted the default setting of chromVAR at 7-mer. I am wondering if 7-mer is sufficient for the research works presented here. Many TFs can recognize much longer binding motifs even over 15nt. Authors need to give solid evidence for the optimization of K-mer signatures.

2. Still the motif. The authors applied TOMTOM to match the de novo motif to the known TFBS. This is totally fine. However, we know many TF will recognize similar TFBS, and it is very difficult to determine the true binding event simply based on the motif match. Even just going through the S1.1, the de novo motif 1 can perfectly match 4 different TFs with slightly different p-value and e-score. Authors have to give extra information and evidence to narrow down the TFs for each motif. Otherwise, if predicted TF does not express in the presented cell types, how the binding could happen?

3. Regarding the TE and ChIP-seq analysis, the overlapping cutoff seems to be just 1bp. This cut-off is really low and should be improved. Moreover, there's no description of the ChIP-seq data used in the study. I guess they used mouse ES ChIP-seq data? The authors need to clearly describe the data and give evidence why the mouse ES ChIP-seq can be used in mouse brain. At least the predicted regions/TEs should be accessible in the mESC. Moreover, when dealing with OCR/ChIP peak overlap, authors need special attention to the cases that the multiple TEs under the same OCR/ChIP peak.

4. For the TE enrichment analysis, the authors did not give enough information of all the overlapping counts but only some enrichment scores. It is difficult to determine the quality especially when the TE subfamily is small.

5. Authors announced MER130 and MamRep434 contain motifs 1 and 3 in the consensus sequence. A sequence-level alignment is necessary to indicate the existence of motif in each TE copy, and the relation between the observed binding and motif should be explored. It will be very powerful if authors can perform an experimental assay, for example, report assay or CRISPR assay to indicate the regulatory potential of specific TE copy.

Minor concern.

1. I noticed authors used 'kmer', which is not very commonly used. I suggest the authors carefully check other studies and use either 'K-mer' or 'k-mer'.
2. please keep the consistency between figures and texts, like de novo should be always italic in Fig1
3. Fig4, Fig5: the description of each axis needs to clarify, and the underline is not recommended.
4. Fig5 c,d: the signal of ChIP-seq input should be added.
5. Fig6, the color needs adjustment. Some points are hard to see.

6. Authors should add the 'data/code availability' section.

Reviewer #3 (Remarks to the Author):

Sekine et al. reanalyze published scATAC-seq data of the mouse brain and indicate a regulatory role of cis-elements located in transposable elements. The authors raise interesting and plausible hypotheses on gene-regulation through these elements. However, the current manuscript only conveys anecdotal support of their hypotheses rather than a comprehensive and robust analysis.

General Comments:

- The manuscript focuses on four TF motifs. What makes these four TFs stand out compared to other known motifs? Why did the authors choose an approach to identify k-mers first and then identify similarities to known TF motifs? I would recommend a more unbiased and comprehensive approach taking into account full databases of TF binding characteristics.
- TF motifs can be ambiguous in the sense that multiple TFs can have highly similar motifs. Motif disambiguation approaches (e.g. PMID:32728250) would be useful in the context of the analyses in the manuscript.
- The manuscript focuses on a select set of TE subfamilies. I would be interested in a more comprehensive survey across all TE subfamilies. How were MER130, MamRep434 selected for follow-up analysis?
- Multiple analyses in the manuscript are likely biased by sequence content. For instance, the enrichment scores in sections 4.4.1 and 4.4.3 do not account for GC or other sequence content. However it is known, that chromatin accessibility can be biased by base composition and tools like chromVAR correct for such biases using background modeling. I do not see this type of correction applied in the current manuscript.
- How do the findings transfer to other dataset? For instance, an analysis of recently published single-cell chromatin accessibility in the developing mouse and human brain would be interesting.
- The code/scripts used for the analysis should be made available.

Specific Comments:

- Section 2.3.1 and Figure 4 left: A simple overlap of TEs with accessibility peaks in specific cell types is likely not robust. Differences in the number of cells in each cell type, the number of peaks should be accounted for. A more fine-scaled characterization taking TE subfamilies into account would be useful.
- Figure 2: for improved readability panels b and c could be aligned.
- Section 2.5 and Figure 7: the absolute numbers of peaks for MER130 and MamRep434 are relatively small. This could result in compromised statistical robustness.
- Section 2.5 and Figure 7: The analysis focuses on bulk ATAC peaks. A more cell type specific analysis would be interesting.
- Section 4.1: How were peaks for each cell type obtained? How were peaks merged into bulk ATAC-seq peaks?

Author response to reviewers' comments

Reviewers' comments:

Reviewer #1 (Remarks to the Author):

In this work, Sekine et al. performed a comprehensive analysis of publicly-available scATAC-seq data from adult mouse cortex to explore cell-type specific cis-elements derived from transposable elements (TEs). The results clearly indicate that MER130 and MamRep434 are contributors that have provided the binding sites of Neurod2 and Lhx2, respectively, in neuronal progenitor cells. The study presents a nice correlation among accessible binding motifs, neuronal cell type, and TEs that contain the binding motifs. The manuscript is well-organized, and the figures and tables are of high quality and easily interpretable. The present report will undoubtedly appeal to wide readers of researchers and is well within the scope of Communications Biology. However, the manuscript lacks a description of the previous finding for putative NeuroD-binding sites derived from MER130. The authors need to address the following points.

1) *The authors should note that a previous study, Notwell et al. (Nat. Comm. 2015, doi: 10.1038/ncomms7644), have also reported about MER130 that contain putative Neurod-binding sites to which p300, an enhancer co-activator, binds in the neocortex of mouse embryos. Although the previous paper has been cited in the manuscript, the authors did not mention that the MER130-derived putative Neurod-binding sequences have been reported previously. Particularly, Notwell et al. (2015) showed that the putative Neurod motif is present in around 230 bp of the MER130 sequence (see Fig. 2 of Notwell's paper), and I do not understand whether or not this is the same as the position of motif1 in 200-300 bp of MER130 reported in the manuscript (Figure 5c and 5d). The authors need to compare the results and add descriptions of their previous findings in the manuscript.*

Response: Thank you for your constructive suggestion. We agree that comparing the previous result of Notwell *et al.* and that of this study is important to clarify our findings regarding the MER130-derived cis-elements. To compare the results, we added several experiments as follows. First, we checked the relative position of the consensus motif (*de novo* motif 1) in the MER130 consensus sequence (Figure 6 b, c). We found that the relative position of *de novo* motif 1 in MER130 was the same as that of the putative Neurod-binding sequence determined previously. Second, we

checked whether accessible motifs in MER130 in the genome correspond to the putative Neurod-binding sites in the consensus MER130 sequence (Figure 6d). We confirmed that the accessible *de novo* motif 1 in MER130 in the genome corresponds to the putative Neurod-binding sequence in the MER130 consensus sequence. Third, we checked whether the motif sequence is conserved among accessible MER130 instances in the genome. We performed multiple alignments of MER130 instances overlapping ATAC peaks and found that *de novo* motif 1 was highly conserved (Figure 6a, Figure S9). This result is consistent with that in the previous study by Notwell *et al.* In addition, to ask to what extent MER130 contributed to the accessible *de novo* motif 1, we checked whether MER130 were enriched with the accessible *de novo* motif 1 compared to all TE populations. We calculated the proportion of TE instances overlapping with accessible *de novo* motif 1 and compared them between MER130 and all TEs (Table 2). We found that MER130 was significantly more enriched with the accessible *de novo* motif 1 than that in all TEs. Fourth, we compared the genomic positions of MER130 instances detected in our result and the previous result (Figure S14). We found that most MER130 instances were common between our and previous results.

Therefore, we concluded that *de novo* motif 1 in MER130 is identical to the previously determined putative binding site of Neurod/g. We also added descriptions of the previous finding in the manuscript and discussed the difference between previous and current results.

In the revised manuscript, this change can be found in the Results: Section 2.3.2 and Discussion.

[Results: Section 2.3.2]

“Notably, our result for MER130 is consistent with that of a previous report showing that MER130 has a putative consensus sequence for Neurod/Neurog binding and functions as an enhancer during mouse neocortical development [23]. To ask whether *de novo* motif 1 in MER130 is the same as the previously determined Neurod/g consensus motif in MER130, we further characterized the *de novo* motif 1 sequence in MER130. We confirmed that the relative position of *de novo* motif 1 in the consensus sequence of MER130 was 216-222, which was the same as the position of the Neurod/g motif determined previously (Figure 6b,c highlighted with orange bar). Then, to confirm whether the MER130-derived *de novo* motif 1 accessible in the mouse genome corresponds to the Neurod/g motif in the consensus sequence of MER130, the accessible motifs detected in the MER130 fragments in the genome were converted into the relative position in the MER130 consensus sequence (the detailed methods are presented in subsection 4.10). The result showed that the accessible motifs were mainly mapped to the putative Neurod/g motif in the MER130 consensus sequence (Figure 6d). Moreover, all MER130 fragments that harbor accessible *de novo* motif 1 contained the putative Neurod/g motif of the MER130 consensus sequence. This result suggests that the motif detection site in the consensus MER130 sequence basically corresponds

to the accessible motif sites in the genomic MER130 fragments and that some MER130 have additional motifs. We further investigated the surrounding sequence of *de novo* motif 1 in the genomic MER130 fragments. We performed multiple alignments of MER130 fragments, which overlap with the ATAC peaks and *de novo* motif 1 derived from the consensus TE sequence (n = 22, the detailed methods are presented in subsection 4.8). The result showed that *de novo* motif 1 was selectively conserved among the accessible MER130 fragments when compared with the flanking regions within MER130 (Figure 6a). Notably, we also observed another highly conserved tandem motif (GGCA and GCCA), which was annotated as a putative Nfi dimer motif in the previous study [23], at an upstream region of *de novo* motif 1 (Figure 6a). Moreover, we aligned all of the accessible MER130 fragments with or without *de novo* motif 1 (n = 26) and confirmed that these motif sites tended to be conserved among these regions (Figure S9). These results suggest that *de novo* motif 1 in MER130 is the same as the previously determined Neurod/g consensus motif in MER130.”

[Results: Section 2.3.2]

“To ask to what extent MER130 and MamRep434 contributed to the accessible *de novo* motifs 1 and 3, respectively, we examined whether these TEs enriched with the accessible *de novo* motifs compared to the whole TE population. We calculated the number of TEs that overlapped or did not overlap with the accessible *de novo* motifs and compared them between each TE (MER130 or MamRep434) and all TEs. The results showed that MER130 and MamRep434 were significantly more enriched with the accessible *de novo* motif 1 and 3, respectively, than all TEs ($q < 0.01$; FDR; Table 2).”

[Discussion]

“Interestingly, a previous report has shown that MER130-derived cis-elements can function as enhancers for NPCs during embryonic neocortical neurogenesis, harboring putative transcription factor binding sites essential for neuronal differentiation, including Neurod/g [23]. In this study, we found that the MER130-derived cis-elements were also active in adult neurogenesis. Indeed, comparing our result of MER130 fragments overlapping ATAC peaks with their results, we were able to cover most of the MER130 fragments they detected with our results (18/23 fragments) and we detected some additional MER130 fragments (18 fragments) with the exact condition of the TE annotation as that found in the previous study (Figure S14). This suggests that many of the MER130-derived cis-elements were commonly accessible both in embryonic and adult neurogenesis, and they might play a fundamental role in neurogenesis. The difference between the data may reflect the different methods for detecting active sites (p300 binding vs scATAC-seq signal) and/or distinct mechanisms between embryonic and adult neurogenesis. Further studies are required to investigate the difference in MER130 function between embryonic and adult neurogenesis. Furthermore, it would be interesting to investigate the differences between other TEs that contribute to the neurogenesis of embryonic and adult mouse brains. Moreover,

comparisons between the brains of multiple species, including mice and humans, would be essential to understand the roles of TEs from the aspect of evolution. These themes will be the next question of this study.”

2) Page 5, line 134–135; “H3K4me1, which is known as an enhancer label”
A reference should be provided in this sentence.

Response: Thank you for your advice. We agree with this comment. Therefore, We have revised and cited [Heintzman, N. D *et al.*, Nat Genet, 2007] for the corresponding sentence in the manuscript (Section 2.3.1) as follows:

“we prepared ChIP peak sets of H3K4me1 and H3K4me3 that are known as enhancer and promoter labels, respectively [47].”

3) Page 8, lines 183–184; “First, we mapped the ChIP-seq reads of transcription factors to the consensus sequences of TEs based on RepeatMasker”

Did the authors map the ChIP-seq reads directly to the consensus sequences of TEs? Or the ChIP-seq reads were mapped to the genome, and then the binding regions were converted to the TE consensus sequences? I think the latter is correct, but this sentence implies the former. Please clarify.

Response: Thank you for your important advice. We apologize for the confusion. First, the “ChIP-seq reads” expression was incorrect. We used ChIP-seq peaks, which had already been processed in the ChIP-Atlas database. We mapped the genome’s ChIP-seq peaks of transcription factors to the consensus sequences of TEs. We revised the sentence in Section 2.4 as follows:

“we mapped the ChIP-seq peaks of transcription factors in the genome to the consensus sequences of TEs based on RepeatMasker [51] (the detailed methods are presented in subsection 4.9).”

We have incorporated a description of the method in Section 4.8 of the Methods. as follows:

“The ChIP-Atlas database provides processed ChIP-seq peaks of multiple experiments [46]. In this experiment, we converted ChIP-seq peaks mapped to the TEs in the genome into the consensus TE sequences.”

We have incorporated your suggestion (“reads” to “peaks”) throughout the manuscript.

4) The authors should state that NDF2 of the HOCOMOCO motif logo in Fig. 2 is the same as Neurod2.

Response: Thank you for your advice. We revised the legend to Figure 2c (Section 2.1).

5) Legend to Figure 2: "De novo motifs"  "De novo motifs"

Response: Thank you for your advice. We revised the legend to Figure 2 (Section 2.1).

6) Font sizes of some figures are too small (e.g., Fig. 2, 3, and 5).

Response: Thank you for your advice. We resized the font in the figures (e.g., Figures 2, 3, 5, and 6).

Reviewer #2 (Remarks to the Author):

Sekine and colleagues presented motif-analysis-based research to understand DNA context within the open chromatin regions (OCR) at the cell-type level. By reanalyzing the scATAC-seq of the mouse brain, the authors confirmed the concordance between OCRs and the K-mers signatures of OCR for the major cell types. They specifically found certain TFBS motifs enriched in different subfamilies of TE, and explored the potential evolution history of these insertion events. The presented results are interesting and solid. The approaches used in the study also reflect how the huge advantage and progress of single-cell experiment technology to facilitate computational data mining in answering biological questions. However, there are certain concerns to the current presented work.

Major concerns.

1. The central approach of this work relies on de novo motif finding. For this step, the authors simply adopted the default setting of chromVAR at 7-mer. I am wondering if 7-mer is sufficient for the research works presented here. Many TFs can recognize much longer binding motifs even over 15nt. Authors need to give solid evidence for the optimization of K-mer signatures.

Response: Thank you for your constructive comment. Although the number varies on the database, one paper suggests that the length of the DNA motif of transcription factors (TFs) ranges from about 5 to 30 base pairs (bp) and around 10 bp on average, and the most common core motif to which TFs bind was 7 bp in eukaryotes [Stewart, AJ *et al.*, Genetics, 2012]. Furthermore, it is known that major TFs important for development share relatively short motifs. For example, basic helix-loop-helix (bHLH) superfamily TFs, including Neurod/g, bind to E-box motifs whose consensus sequence is ‘CANNTG’ [Susan, Genome Biol, 2004], and the homeodomain TFs, including HOX genes, resemble consensus motif ‘(C/G)TAATT(G/A)’ [Noyes *et al.*, Cell, 2008]. Therefore, we used the motif length of 7-mer. Our analysis revealed that these short motifs were detected as variably accessible motifs, and some were essential for TE regulation in the neural cell population. To confirm the consistency of the result, we checked k-mers around 7-mer for the motif detection process in chromVAR. When k was no more than 5, no motifs were generated. When k was 6 or 8, we observed similar seed k-mers with high variability scores to those of 7 (*de novo* motifs 1-4) with a default threshold parameter (1.5) as follows:

7-mer: [CAGATGG, CACCCAC, CTAATTA, ACAGCTG]

6-mer: [CAGATG, ACCCAC, TAATTA, CAGCTG]

8-mer: [ACAGATGG, CACCCACA, CTAATTA, CACAGCTG]

Also, when k was significant (e.g., 15), the calculation could not be performed in a realistic amount of time. Thus, we concluded that the 7-mer motif is appropriate for this paper. In the revised manuscript, this change can be found in the Methods (Section 4.2.1) as follows:

“In eukaryotes, the most common core motif to which transcription factors bind is 7 bp in length [64], therefore, we chose the 7-mer in this study, which is also the default length for using chromVAR [29] in the STREAM paper [40].”

Comment 2. Still the motif. The authors applied TOMTOM to match the *de novo* motif to the known TFBS. This is totally fine. However, we know many TF will recognize similar TFBS, and it is very difficult to determine the true binding event simply based on the motif match. Even just going through the S1.1, the *de novo* motif 1 can perfectly match 4 different TFs with slightly different p-value and e-score. Authors have to give extra information and evidence to narrow down the TFs for each motif. Otherwise, if predicted TF does not express in the presented cell types, how the binding could happen?

Response: Thank you for your constructive suggestion. We agree that only the sequence motif cannot determine which transcription factors can bind to the motif. To narrow down the candidate transcription factors for each motif, we examined RNA exp

ression levels of the candidate transcription factors in each cell population. We transferred scRNA-seq data from other data sets of the adult mouse cortex based on each cell type label using Seurat (Figure 2d). Then we estimated which transcription factor could contribute to the accessible *de novo* motifs based on their expression levels. We also calculated gene activity levels from the chromatin accessibility of each gene using scATAC-seq data to confirm the consistency of the scRNA-seq results for RNA expression (Figure S4). We obtained the same conclusion as that in our original manuscript that suggests *Neurod2* and *Lhx2* contributed most to *de novo* motifs 1 and 3, respectively. However, we found that *Gli2*, the candidate transcription factor for *de novo* motif 2, was not expressed as much in the excitatory neurons where *de novo* motif 2 was highly accessible. Instead, *Egr3* was the most expressed gene among candidate transcription factors of *de novo* motif 2. Similarly, we found that *Meis2*, the candidate transcription factor for *de novo* motif 4, was not expressed as much in the inhibitory neurons and interneurons where *de novo* motif 4 was highly accessible. Instead, *Tcf4* was the most expressed gene among candidate transcription factors of *de novo* motif 4. According to these results, we revised the corresponding parts of the manuscript and figures (Section 2.1 and Figure 2). We also removed the description regarding *Gli2* and *Meis2*. Moreover, along with this revision, we improved our analysis for the accessible motifs and their corresponding cell types and added some supplemental figures (Figures S1, S2, S3).

In the revised manuscript, this change can be found in the Results [Section 2.1] as follows:

“To check to what extent the candidate transcription factor genes are actively transcribed in the cell types where the *de novo* motifs are accessible, we examined their RNA expression levels using public scRNA-seq data of the adult mouse cortex [34] and gene activity levels using scATAC-seq data [28] (the detailed methods are presented in subsection 4.1, 4.3). We confirmed that specific transcription factors of the candidates that regulate neural differentiation and/or neuronal activity are highly expressed in each cell population [4, 6, 35, 36] (Figure 2d, Figure S4). In the cells where *de novo* motif 1 was highly accessible (that is, excitatory neurons), the results showed that *Neurod2* was expressed the most compared to that of other candidates of transcription factors, including *Neurog2*, *Neurod1*, *Atoh1*, and *Olig2*. Therefore, *Neurod2* was assumed to be the factor that contributed most to *de novo* motif 1 accessibility. Similarly, *Egr3* for *de novo* motif 2 (accessible in excitatory neurons), *Lhx2* for *de novo* motif 3 (accessible in excitatory neurons and astrocytes), and *Tcf4* for *de novo* motif 4 (accessible in interneurons and inhibitory neurons) were estimated to be transcription factors that contribute the most to the *de novo* motif accessibility (Figure 2c). Notably, *de novo* motif 3, which corresponds to putative *Lhx2* binding sites, was highly accessible across a part of excitatory neurons and astrocytes (Figure 2b,d). This suggests that *Lhx2* regulation is important for not only neurons but also astrocytes, consistent with the results

of a recent report showing that Lhx2 plays an essential role in astrocyte maturity through transcriptional and chromatin regulation [37].”

3. Regarding the TE and ChIP-seq analysis, the overlapping cutoff seems to be just 1bp. This cut-off is really low and should be improved. Moreover, there’s no description of the ChIP-seq data used in the study. I guess they used mouse ES ChIP-seq data? The authors need to clearly describe the data and give evidence why the mouse ES ChIP-seq can be used in mouse brain. At least the predicted regions/TEs should be accessible in the mESC. Moreover, when dealing with OCR/ChIP peak overlap, authors need special attention to the cases that the multiple TEs under the same OCR/ChIP peak.

Response: Thank you for your suggestions. Regarding the overlapping cutoff of the TE and ChIP-seq peak, 1bp was appropriate because the enrichment score can quantitatively evaluate the degree of overlap between the TE and ChIP peak. Namely, the TE, with less overlap with the ChIP peak, has a relatively low score (Section 4.6.1 (1)). Even when the calculation of the enrichment score was restricted to TEs overlapping with the ChIP peak center (i.e., cutoff changed to over half of the ChIP peak width), there was no change in the TE of the top scores such as MER130 and Mam Rep434. In cases where multiple TEs overlap with the same ChIP peak, the overlap between each TE and the peak was separately calculated in base pair (bp) units using the intersectBed tool. In the revised manuscript, this change can be found in Section 4.6.1 as follows:

“The enrichment score quantitatively evaluated the degree of overlap between the TE and ChIP peak; TEs with less overlap with ChIP peaks have a relatively low score. In cases where multiple TEs overlap with the same ChIP peak, the overlap between each TE and the peak was separately calculated in base pair (bp) units using the intersectBed tool.”

Regarding the overlap between TEs and ATAC peaks with motifs, we counted the number of ATAC peaks containing the entire motif, which is included in both TE and the ATAC peak. Thus, if multiple TEs overlapped with the same ATAC peak, only the TE containing the entire motif was counted. We apologize for the incorrect description, and in the revised manuscript, this change can be found in Section 4.6.3 as follows:

“The intersectBed tool in the BEDTools package [65] was used to obtain the ATAC peaks where the entire motif was detected. Subsequently, as “number of ATAC peaks with motifs in the TE subfamily” in (2), we counted the number of ATAC peaks containing the entire motif, which is included in both the TE and the ATAC peaks.”

We utilized the ChIP-Atlas database for the ChIP-seq data, and the basic description was included in the Materials and Methods (Section 4.1). Of the data available in ChIP-Atlas, we used only the neural cell type class after P0. We added more detailed descriptions for ChIP-seq data in each experiment, and in the revised manuscript, this change can be found in the Result for H3K4me1/3: Section 2.3.1, Result for Neurod2 and Lhx2: Section 2.4, and Materials and Methods: Section 4.1 as follows:

[Result for H3K4me1/3: Section 2.3.1]

“From the ChIP-Atlas database [46], we prepared ChIP peak sets of H3K4me1 and H3K4me3 that are known as enhancer and promoter labels, respectively [47].”

[Result for Neurod2 and Lhx2: Section 2.4]

“We used ChIP-seq data of Neurod2 and Lhx2 in the neural cell type after P0 in the ChIP-Atlas database [46].”

[Materials and Method: Section 4.1]

“Of the data available in the ChIP-Atlas, we used only the neural cell type class after P0.”

4. For the TE enrichment analysis, the authors did not give enough information of all the overlapping counts but only some enrichment scores. It is different to determine the quality especially when the TE subfamily is small.

Response: Thank you for your constructive advice. For the enrichment score, we added information on the numerator and denominator values used in the calculation, such as the number of overlaps between the ATAC peaks and the TE subfamily (Supplementary Data 1). To prevent the enrichment score from being abnormally high due to a small number of TE fragments, we restricted the calculation to the subfamilies where ATAC peaks and motif-containing TEs intersected at least three times in the whole genome (Section 4.6.3). However, there are still cases where the overlap between TE and ATAC peaks containing *de novo* motifs was small. Therefore, we added Fisher’s exact test with Benjamini-Hochberg correction for statistical significance based on the numbers used to calculate the enrichment scores (Supplementary Data 1). We confirmed that each TE described in the manuscript (MER130, MamRep137, MER102b, MamRep434, LFSINE_Vert) was significantly enriched in both the accessible *de novo* motifs and the ChIP-seq peaks ($q < 0.05$: false discovery rate (FDR)). In the revised manuscript, this change can be found in the Results: Section 2.3.1 as follows:

“As a result, MER130 (unknown class in Repbase [48] but currently considered DNA class [49]) was the most enriched TE subfamily in Neurod2 binding peaks and *de novo* motif 1 ($q < 0.01$: false discovery rate (FDR)) (Figure 5a, Supplementary Data 1). MER124 (DNA class in Repbase [48]) and MamRep434 (DNA class in Repbase [48]) were the most enriched TE subfamilies in Lhx2 binding peaks and *de novo* motif 3, respectively ($q < 0.01$: FDR) (Figure 5a, Supplementary Data 1).”

5. Authors announced MER130 and MamRep434 contain motifs 1 and 3 in the consensus sequence. A sequence-level alignment is necessary to indicate the existence of motif in each TE copy, and the relation between the observed binding and motif should be explored. It will be very powerful if authors can perform an experimental assay, for example, report assay or CRISPR assay to indicate the regulatory potential of specific TE copy.

Response: Thank you for your constructive suggestion. We performed multiple sequence alignments of MER130 fragments overlapping with the ATAC-seq peak and *de novo* motif 1 ($n = 22$). We confirmed that the genomic MER130 fragments which met the condition above had the exact *de novo* motif 1 (CAGATGG) derived from the consensus sequence (Figure 6a (1)). Moreover, mapping of Neurod2 ChIP-seq peaks to the MER130 consensus sequence revealed that Neurod2 binding occurred surrounding the *de novo* motif 1 (Figure 6b,c left). Similarly, we also performed the same experiments for MamRep434. We aligned MamRep434 fragments overlapping with the ATAC-seq peak and *de novo* motif 3 ($n = 8$). We confirmed that 5 out of 8 MamRep434 fragments had the exact *de novo* motif 3 (CTAATTA) derived from the consensus *de novo* motif 3 (Figure 6a (2)). In addition, Lhx2 ChIP-seq peaks were mapped surrounding this position, suggesting that Lhx2 binding occurred via this motif (Figure 6b,c right). These results indicate that *de novo* motifs 1 and 3 in the MER130 and MamRep434 consensus sequences exist in the TE copies in the genome and could function as consensus motifs for Neurod2 and Lhx2 binding, respectively.

Unfortunately, we do not have an experimental environment, and we cannot perform any assay to confirm the regulatory potential of the TE copies. Such experiments are interesting, and we hope future studies will examine them. In the revised manuscript, this change can be found in the Results: Section 2.3.2 as follows:

“We performed multiple alignments of MER130 fragments, which overlap with the ATAC peaks and *de novo* motif 1 derived from the consensus TE sequence ($n = 22$, the detailed methods are presented in subsection 4.8). The result showed that *de novo* motif 1 was selectively conserved among the accessible MER130 fragments when compared with the flanking regions within MER130 (Figure 6a). Notably, we also observed another highly conserved tandem motif (GGCA and GCCA), which was annotated as a put

active Nfi dimer motif in the previous study [23], at an upstream region of *de novo* motif 1 (Figure 6a). Moreover, we aligned all of the accessible MER130 fragments with or without *de novo* motif 1 (n = 26) and confirmed that these motif sites tended to be conserved among these regions (Figure S9). These results suggest that *de novo* motif 1 in MER130 is the same as the previously determined Neurod/g consensus motif in MER130. Similarly, we also performed these experiments to characterize *de novo* motif 3 in MamRep434. We found that *de novo* motif 3 tended to be conserved among accessible MamRep434 fragments (Figure 6a, Figure S9), and confirmed that accessible motifs in the genomic MamRep434 fragments were mainly mapped to the *de novo* motif 3 (Lhx2-like motif) in the MamRep434 consensus sequence (Figure 6d right panel).”

Minor concern.

1. I noticed authors used 'kmer', which is not very commonly used. I suggest the authors carefully check other studies and use either 'K-mer' or 'k-mer'.

Response: Thank you for your advice. We have revised it.

2. please keep the consistency between figures and texts, like *de novo* should be always italic in Fig1

Response: Thank you for your advice. We have revised it.

3. Fig4, Fig5: the description of each axis needs to clarify, and the underline is not recommended.

Response: Thank you for your advice. We have revised the axis labels of Figures 4a and b in Section 2.3.1 and Figure 5a in Section 2.3.2.

4. Fig5 c,d: the signal of ChIP-seq input should be added.

Response: Thank you for your advice. To account for the effect of input bias of ChIP-seq, we additionally evaluated the IP/input ratio for the proportion of reads that mapped to the detection sites of the TE-derived *de novo* motif. The IP/input ratios for both *de novo* motif 1 in MER130 and *de novo* motif 3 in MamRep434 exceeded over 2-fold, suggesting that the transcription factors bindings were enriched at t

he detection sites of the motifs (Section 2.4 Figure S10). In the revised manuscript, this change can be found in the Results: Section 2.4 as follows:

“To account for the effect of input bias of ChIP-seq, we also evaluated the IP/input ratio for the proportion of reads that mapped to the detection sites of the TE-derived *de novo* motifs (the detailed methods are presented in subsection 4.9). The IP/input ratios for *de novo* motif 1 in MER130 and *de novo* motif 3 in MamRep434 exceeded over 2-fold, suggesting that the transcription factor bindings were enriched at the detection sites of the motifs (Figure S10). These results suggest that the *de novo* motifs in these TEs play important roles as transcription factor-binding sites.”

5. Fig6, the color needs adjustment. Some points are hard to see.

Response: Thank you for your advice. We have revised it in the manuscript. This change can be found in Section 2.4, Figure 7a.

6. Authors should add the 'data/code availability' section.

Response: Thank you for your advice. We have added the 'Data/code availability' section.

Reviewer #3 (Remarks to the Author):

Sekine et al. reanalyze published scATAC-seq data of the mouse brain and indicate a regulatory role of cis-elements located in transposable elements. The authors raise interesting and plausible hypotheses on gene-regulation through these elements. However, the current manuscript only conveys anecdotal support of their hypotheses rather than a comprehensive and robust analysis.

General Comments:

1. The manuscript focuses on four TF motifs. What makes these four TFs stand out compared to other known motifs? Why did the authors choose an approach to identify k-mers first and then identify similarities to known TF motifs? I would recommend a more unbiased and comprehensive approach taking into account full databases of TF binding characteristics.

Response: Thank you for your interesting suggestion. First, we would like to clarify the purpose of this study. In this study, we aimed to find essential transposable elements (TEs) that contribute to not only specific transcription factor (TF) binding but also cell-type specific regulation in the mouse adult brain at single-cell levels (Figure 1). To this end, we first investigated the most variably accessible motifs of cis-elements across cell types, which could characterize specific cell populations at single-cell levels using scATAC-seq data.

Because we considered that some TE-derived cis-elements could be regulated in unknown populations that are not characterized enough, we preferred an unbiased method that does not depend on known cell types. Therefore, we used scATAC-seq data to describe unbiased cell populations and their regulatory cis-elements. Indeed, it is a fascinating approach to use ChIP-seq data of specific populations in the mouse adult brain first and then analyze for related TEs. However, in that case, target cell populations for analysis are restricted to only known bulk populations. Furthermore, ChIP-seq data of specific cell types in the mouse adult brain are limited. In addition, at the beginning of this study, there was little data available for TF binding profile at single-cell resolution. Thus, we chose a strategy to utilize scATAC-seq data, which can reveal accessible motifs at single-cell levels, and then estimate putative binding TFs to these motifs.

Accordingly, we analyzed the most variably accessible motifs, which were regulated in a cell-type or cell-population-dependent manner using scATAC-seq data of the mouse adult brain with chromVAR (Figure 2). Consequently, we got the most highly regulated four *de novo* motifs as candidates using an arbitrary threshold set for $r > 2.0$ (Section 3). Importantly, subsequent analysis revealed that two of these motifs, *de novo* motif 1 and *de novo* motif 3, were significantly regulated in a putative neuronal progenitor cell population, which was not labeled in the original scATAC-seq data (Figure 3). Although this population was still not characterized well in the previous report [Cusanovich *et al.*, Cell, 2018], we speculate that this population consisted of neuroblasts migrating from SVZ or other neuronal progenitors with unknown origins. Then we focused on these motifs and found that two TEs, MER130 and MamRep434, were enriched in *de novo* motifs 1 and 3, respectively, and contributed to specific TF binding (Figures 5 and 6). Moreover, our results suggest that MER130- and MamRep434-derived cis-elements were significantly regulated in putative neuronal progenitor cells in the mouse adult brain (Figure 5).

Thus, our approach suited our purpose and enabled us to find unique TEs contributing to cell-type-specific regulation, especially in the context of adult neurogenesis. We revised the Discussion part of the manuscript to add the overview and discussion described above. We also rearranged the supplemental and main figures into one figure (Figure 5) to improve readability. In the revised manuscript, this change can be found in the Discussion Section as follows:

“Accumulating evidence from recent papers suggests that TEs are one of the important sources of enhancers and contribute to the evolution of species, conferring ne

w regulatory networks of transcription factors [18, 22, 58]. Based on this, we hypothesized that during evolution, some TEs contribute to having specific cell populations or regulating the function of particular populations in the mouse brain (Figure 1). Thus, in this study, we aimed to find important TEs that contribute to cell-type specific regulation in the mouse adult brain. To this end, we first investigated the most variably accessible motifs of cis-elements across cell types, which could characterize specific cell populations at single-cell levels using scATAC-seq data. Because we considered that some TE-derived cis-elements could be regulated in unknown populations that are not characterized enough, we preferred an unbiased method that does not depend on known cell types. Therefore, we used scATAC-seq data to characterize unbiased cell populations and their regulatory cis-elements. Accordingly, we analyzed the most variably accessible motifs, which were regulated in a cell-type or cell-population-dependent manner using scATAC-seq data of the mouse adult brain with chromVAR [29] (Figure 2). Consequently, we obtained the most variable four *de novo* motifs as candidates. Interestingly, pseudo-time analysis showed that *de novo* motif 1 (Neurod2-like motif) and *de novo* motif 3 (Lhx2-like motif) were accessible in the branch of putative glutamatergic neuronal progenitor cells, especially those suggested to be related to stress-responsive neurogenesis for tissue recovery (subsection 2.2). In fact, Neurod2 and Lhx2 have been reported to function in the neurogenesis of glutamatergic neurons [44, 45]. In addition, Neurod2 has been proposed to function as a nexus for stress-related dendritic synaptic plasticity [44]. This suggests that *de novo* motif 1 and *de novo* motif 3 were regulated in a putative neuronal progenitor cell population, which was not labeled in the original scATAC-seq data. Although this population was still not characterized well in the previous report [28], we speculate that this population consisted of neuroblasts migrating from SVZ or other neuronal progenitors with unknown origins. Subsequently, we focused on these motifs and found that two TEs, MER130 and MamRep434, were enriched in *de novo* motif 1 and *de novo* motif 3, respectively. In addition, they were enriched in transcription factor binding sites of Neurod2 and Lhx2, respectively (Figure 5). The results of ChIP-seq peaks mapping to consensus sequences of the TEs suggest that MER130 and MamRep434 induce Neurod2 and Lhx2 binding based on their internal motifs (subsubsection 2.3.1). In addition, MER130 and MamRep434 TEs themselves were significantly accessible in a putative neuronal progenitor cell population (subsubsection 2.3.2), suggesting that MER130- and MamRep434-derived cis-elements serve as open chromatin for the binding of Neurod2 and Lhx2 transcription factors, especially in neuronal progenitors in the mouse adult brain (Figure 5). Thus, our approach has enabled us to find unique TEs that contribute to cell-type-specific regulation, especially in the context of adult neurogenesis.”

2. TF motifs can be ambiguous in the sense that multiple TFs can have highly similar motifs. Motif disambiguation approaches (e.g. PMID:32728250) would be useful in the context of the analyses in the manuscript.

Response: Thank you for your constructive suggestion. We agree that only the sequence motif is insufficient to determine which transcription factors can bind to the motif. This was pointed out by another reviewer too. We considered using the suggested data, but we noticed it was for the human TFs and motifs and could not directly apply it to the mouse data we used. Therefore, we considered using an alternative approach. To narrow down the candidate transcription factors for each motif, we examined RNA expression levels of the candidate transcription factors in each cell population. We transferred scRNA-seq data from other data sets of the adult mouse cortex based on each cell type label using Seurat (Figure 2d). Then we estimated which transcription factor could contribute to the accessible *de novo* motifs based on their expression levels. We also calculated gene activity levels from the chromatin accessibility of each gene using scATAC-seq data to confirm the consistency of the scRNA-seq results for RNA expression (Figure S4). We obtained the same conclusion as that in our original manuscript that suggests *Neurod2* and *Lhx2* contributed most to *de novo* motifs 1 and 3, respectively. However, we found that *Gli2*, the candidate transcription factor for *de novo* motif 2, was not expressed as much in the excitatory neurons where *de novo* motif 2 was highly accessible. Instead, *Egr3* was the most expressed gene among candidate transcription factors of *de novo* motif 2. Similarly, we found that *Meis2*, the candidate transcription factor for *de novo* motif 4, was not expressed as much in the inhibitory neurons and interneurons where *de novo* motif 4 was highly accessible. Instead, *Tcf4* was the most expressed gene among candidate transcription factors of *de novo* motif 4. According to these results, we revised the corresponding parts of the manuscript and figures (Section 2.1 and Figure 2). We also removed the description regarding *Gli2* and *Meis2*. Moreover, along with this revision, we improved our analysis for the accessible motifs and their corresponding cell types and added some supplemental figures (Figure S1, S2, S3). In the revised manuscript, this change can be found in the Results: Section 2.1 as follows:

“To check to what extent the candidate transcription factor genes are actively transcribed in the cell types where the *de novo* motifs are accessible, we examined their RNA expression levels using public scRNA-seq data of the adult mouse cortex [34] and gene activity levels using scATAC-seq data [28] (the detailed methods are presented in subsection 4.1, 4.3). We confirmed that specific transcription factors of the candidates that regulate neural differentiation and/or neuronal activity are highly expressed in each cell population [4, 6, 35, 36] (Figure 2d, Figure S4). In the cells where *de novo* motif 1 was highly accessible (that is, excitatory neurons), the results showed that *Neurod2* was expressed the most compared to that

of other candidates of transcription factors, including *Neurog2*, *Neurod1*, *Atoh1*, and *Olig2*. Therefore, *Neurod2* was assumed to be the factor that contributed most to *de novo* motif 1 accessibility. Similarly, *Egr3* for *de novo* motif 2 (accessible in excitatory neurons), *Lhx2* for *de novo* motif 3 (accessible in excitatory neurons and astrocytes), and *Tcf4* for *de novo* motif 4 (accessible in interneurons and inhibitory neurons) were estimated to be transcription factors that contribute the most to the *de novo* motif accessibility (Figure 2c). Notably, *de novo* motif 3, which corresponds to putative *Lhx2* binding sites, was highly accessible across a part of excitatory neurons and astrocytes (Figure 2b,d). This suggests that *Lhx2* regulation is important for not only neurons but also astrocytes, consistent with the results of a recent report showing that *Lhx2* plays an essential role in astrocyte maturity through transcriptional and chromatin regulation [37].”

3. The manuscript focuses on a select set of TE subfamilies. I would be interested in a more comprehensive survey across all TE subfamilies. How were MER130, MamRep434 selected for follow-up analysis?

Response: Thank you for your constructive comment. We selected MER130 and MamRep434 for detailed analysis for the following reasons. First, because this study aims to find the TEs that regulate specific cell populations, we focused on the TEs enriched in the specific pseudo-time branch among TEs with accessible TF binding motifs. MER130 and MamRep434 were enriched in a particular branch (Figure 5), and this branch was characterized as a putative neuronal progenitor cell population (Figure 3, S7). In addition, MER130 and MamRep434 had the highest enrichment scores for *de novo* motifs 1 and 3, respectively (Figure 5a). Second, because we hypothesized that TEs originally having TF binding motifs could primarily contribute to the cis-elements, we focused on TEs that contain TF motifs in their consensus sequences. We found that MER130 and MamRep434 had the consensus motifs of TF binding in their consensus sequences (Figure 6). Therefore, we decided to focus on these two TEs and perform subsequent analysis. We added an explanation to clarify the reason for the follow-up analysis. In the revised manuscript, this change can be found in the Results: Section 2.3.2 as follows:

“From the results described above, we found that MER130 and MamRep434 had the highest enrichment scores for *de novo* motifs 1 and 3, respectively, and were significantly regulated in the putative neuronal progenitors (Figure 5a,d). In addition, the *de novo* motifs were observed in their consensus sequences (Figure 6). These results prompted us to investigate these two TEs in detail for further analysis.”

Still, we would also be interested in the other TE subfamilies that Neurod2 and Lhx2 regulate in the putative neuronal progenitors. Notably, three subfamilies (MamRep137 and MER102b for Neurod2 binding and LFSINE_Vert for Lhx2 binding) were also identified as the branch-specific accessible TE candidates enriched for TF binding and accessible *de novo* motifs (Figure 5a red triangles). However, we could not detect TF motifs in their consensus sequences. Therefore, we added further analysis for these TE subfamilies to investigate whether these TEs could function as cis-elements. First, we converted ChIP-seq data of Neurod2 or Lhx2 to these TE consensus sequences and found that the ChIP-seq peaks were enriched in each TE consensus sequence (Figure S11(1) and (2)). This result suggests that these transcription factors can bind to each TE. Then, to compare the ChIP-seq data and the relative position of the accessible *de novo* motifs (TF binding motifs), we mapped the accessible motifs in the genomic TEs to the TE consensus sequences (Figure S11 (3)). We found that the accessible motifs of the genomic MamRep137 or LFSINE_Vert were mainly mapped to one position near the center of the ChIP-seq peak in the consensus sequence. In contrast, accessible motifs of the genomic MER102b were mapped to multiple sites around the ChIP-seq peak. These results suggest that these TEs also could contribute to the transcription factor binding sites through the *de novo* motifs. Given that these motifs were not detected in the TE consensus sequences, these TEs might acquire transcription factor-binding sites by accumulating pre-motif mutations in each site during evolution rather than having an original consensus motif for TF binding. This notion is consistent with a recent paper suggesting that independent TEs can evolve into enhancers with the same function through convergent evolution, especially observed in redundant enhancers [Barth *et al.*, Genome Biol Evol, 2020]. This finding suggests that the transcriptional network is formed not only in the TE groups that are considered to have originally had motifs, but also in other TE groups that are considered to have later acquired motifs during evolution. In the revised manuscript, this change can be found in the Results and Discussion Sections.

[Result: Section 2.4]

“We also checked whether other TEs identified as the candidates for transcription factor binding in the putative neuronal progenitors could function as cis-elements (that is, MamRep137 and MER102b for Neurod2 binding, and LFSINE Vert for Lhx2 binding) (Figure 5a red triangles). We converted the ChIP-seq data of Neurod2 or Lhx2 to the TE consensus sequences and found that the ChIP-seq peaks were enriched in each TE consensus sequence (Figure S11(1), (2)). This result suggests that these transcription factors can bind to each TE subfamily. Then, to compare the ChIP-seq data and the relative position of the accessible *de novo* motifs (transcription factor binding motifs), we mapped the accessible motifs in the genomic TEs to the TE consensus sequences (Figure S11(3)). We found that the accessible motifs of the genomic MamRep137 or LFSINE Vert were mainly mapped to one position near the center of the ChIP-seq peak in the consensus sequence. In contrast, accessible motifs of the

genomic MER102b were mapped to multiple sites around the ChIP-seq peak. These results suggest that these TEs also could contribute to the transcription factor binding sites through the *de novo* motifs.”

[Discussion]

“Importantly, we found that the transcription factor binding motifs (that is, Neurod2 motif in MER130 and the Lhx2 motif in MamRep434) tended to be conserved not only among the TE subfamily accessible in the mouse genome but also among that of mammalian species (Figure 6a, Figure S9, Figure S13). Given that these motifs are also observed in the TE consensus sequences, these TEs could originally have had consensus motifs for transcription factors, and the motifs could have been protected from gaining mutations during evolution through natural selection. In contrast, we could not detect transcription factor motifs in the TE consensus sequences of MamRep137 and MER102b, which were enriched for the Neurod2 transcription factor, and LFSINE Vert which was enriched for the Lhx2 transcription factor (Figure S11). These TEs might acquire transcription factor-binding sites by accumulating mutations in pre-motifs in each site during evolution. This notion is consistent with that in a recent paper suggesting that independent TEs can evolve into enhancers with the same function through convergent evolution, mainly observed in redundant enhancers [59]. This finding indicates that the transcriptional network is formed not only in the TE groups that are considered to have originally had motifs but also in other TE groups that are considered to have later acquired motifs during evolution.”

4. Multiple analyses in the manuscript are likely biased by sequence content. For instance, the enrichment scores in sections 4.4.1 and 4.4.3 do not account for GC or other sequence content. However it is known, that chromatin accessibility can be biased by base composition and tools like chromVAR correct for such biases using background modeling. I do not see this type of correction applied in the current manuscript.

Response: Thank you for your constructive suggestion. To account for technical biases (GC content and average accessibility) of scATAC-seq peaks, we additionally calculated the z-score using the enrichment score of accessible motifs based on the chromVAR’s background peak set as a control (the detailed methods are presented in Subsubsection 4.6.3). As a result, we confirmed that accessible *de novo* motif 1 and *de novo* motif 3 were enriched in MER130 and MamRep434, respectively (Section 2.3.1, Supplementary Data 1). For the enrichment score, we added information on th

e numerator and denominator values used in the calculation, such as the number of overlaps with the ATAC peaks of the TE subfamily (Supplementary Data 1). In the revised manuscript, this change can be found in the Results: Section 2.3.1. as follows:

“To account for technical biases (GC content and average accessibility) of scATAC-seq peaks, we calculated the z-score using the enrichment score of accessible motifs based on the chromVAR’s background peak set as a control (the detailed methods are presented in subsection 4.6.3). As a result, we confirmed that MER130 and MamRep434 were enriched in *de novo* motifs 1 and 3, respectively (z-score>2; Supplementary Data 1). These results suggest that specific TE subfamilies significantly contribute to Neurod2 and Lhx2 binding sites.”

5. How do the findings transfer to other dataset? For instance, an analysis of recently published single-cell chromatin accessibility in the developing mouse and human brain would be interesting.

Thank you for your interesting suggestion. We agree that applying our pipeline to various scATAC-seq data and comparing them is interesting. Consistent with a previous paper suggesting that MER130 was important for embryonic neurogenesis [Notwell *et al.*, Nat Commun, 2015], one of the important findings of this study is that MER130 was also important for adult neurogenesis (Figure 5). Therefore, comparing the TEs contributing to the neurogenesis between the embryonic and adult mouse brain would be interesting. Moreover, comparisons between multiple species of the brain, including mice and humans, would be essential to understand the roles of TEs from the aspect of evolution. These themes will be the next question of this study. However, this is beyond the scope of the current study, and I would like to investigate it in future work. We added a discussion regarding MER130’s function in embryonic and adult neurogenesis and the future direction of the study. In the revised manuscript, this change can be found in the Discussion Section as follows:

“Interestingly, a previous report has shown that MER130-derived cis-elements can function as enhancers for NPCs during embryonic neocortical neurogenesis, harboring putative transcription factor binding sites essential for neuronal differentiation, including Neurod/g [23]. In this study, we found that the MER130-derived cis-elements were also active in adult neurogenesis. Indeed, comparing our result of MER130 fragments overlapping ATAC peaks with their results, we were able to cover most of the MER130 fragments they detected with our results (18/23 fragments) and we detected some additional MER130 fragments (18 fragments) with the exact condition of the TE annotation as that found in the previous study (Figure S14). This suggests that many of the MER130-derived cis-elements were commonly accessible both in embryonic and adult neurogenesis, and they might play a fundamental role in neuroge

nesis. The difference between the data may reflect the different methods for detecting active sites (p300 binding vs scATAC-seq signal) and/or distinct mechanisms between embryonic and adult neurogenesis. Further studies are required to investigate the difference in MER130 function between embryonic and adult neurogenesis. Furthermore, it would be interesting to investigate the differences between other TEs that contribute to the neurogenesis of embryonic and adult mouse brains. Moreover, comparisons between the brains of multiple species, including mice and humans, would be essential to understand the roles of TEs from the aspect of evolution. These themes will be the next question of this study.”

6. The code/scripts used for the analysis should be made available.

Response: Thank you for your comment. We added the ‘Data/code availability’ section.

Specific Comments:

1. Section 2.3.1 and Figure 4 left: A simple overlap of TEs with accessibility peaks in specific cell types is likely not robust. Differences in the number of cells in each cell type, the number of peaks should be accounted for. A more fine-scaled characterization taking TE subfamilies into account would be useful.

Response: Thank you for your constructive comment. We agree that differences in the number of cells in each cell type can affect the quality of pseudo-bulk ATAC peaks and, consequently, the number of accessible TEs. Therefore, we subsampled the same number of cells for each cell type to match the minimum number among the cell types (the minimum was 197 cells of microglia). Then we merged subsampled scATAC peaks to make pseudo-bulk ATAC peaks for each cell type and computed the proportion of TEs in the ATAC peaks. To account for the differences in the number of pseudo-bulk ATAC peaks of each cell type, we performed Fisher’s exact test for statistical significance (Table S1). We obtained a consistent result to our original data and concluded that TEs were more accessible in the excitatory neurons than in other cell types. In the revised manuscript, this change can be found in the Results: Section 2.3.1 as follows:

“To account for the differences in the number of cells of each cell type that can affect the quality of pseudo-bulk ATAC peaks, we subsampled the same number of cells for each cell type and compared the proportion of the peaks overlapping with TEs. We obtained a consistent result with data that used the whole cell population and concluded that TEs were significantly more accessible in the excitatory neurons than in other cell types ($q < 0.01$: Fisher’s exact test with the Benjamini-Hochberg correction; Table S1).”

2. Figure 2: for improved readability panels b and c could be aligned.

Response: Thank you for your advice. We have revised it (Section 2.1, Figure 2).

3. Section 2.5 and Figure 7: the absolute numbers of peaks for MER130 and MamRep434 are relatively small. This could result in compromised statistical robustness.

Response: Thank you for your advice. We additionally tested for the independence of the acquisition times of these TE-derived cis-elements using Fisher's exact test. A comparison of the number of TE-derived cis-elements acquired in the ancestors of Amniota and Eutheria showed that they were significantly different between MER130 and MamRep434 ($p < 0.01$, Fisher's exact test). In the revised manuscript, this change can be found in the Results: Section 2.5 as follows:

“We performed Fisher's exact test to ask whether the acquisition time of these TE-derived cis-elements was independent. A comparison of the number of TE-derived cis-elements acquired in the ancestors of Amniota, and Eutheria showed that they significantly differed between MER130 and MamRep434 ($p < 0.01$, Fisher's exact test).”

4. Section 2.5 and Figure 7: The analysis focuses on bulk ATAC peaks. A more cell type specific analysis would be interesting.

Response: Thank you for your interesting suggestion. We performed additional experiments with ATAC-seq peaks for each cell type. The results were also similar for each cell type and differentiation process of the excitatory neurons (Section 2.5, Figure S12) and complemented our result. In the revised manuscript, this change can be found in the Results: Section 2.5 as follows:

“These results were also similar for each cell type and differentiation process of the excitatory neurons (Figure S12), suggesting that MER130- and MamRep434-derived cis-elements were acquired mainly in the ancestors of Amniota and Eutheria, respectively.”

5. Section 4.1: How were peaks for each cell type obtained? How were peaks merged into bulk ATAC-seq peaks?

Response: Thank you for your advice. In the revised manuscript, we added a description for the peak calling method (Section 4.1) as follows:

“We first split the scATAC-seq data into bam files for each cell type label and performed peak calling using MACS2 [63] with the following parameters: “- nonmodel -keep-dup all-extsize 200-shift -100” for each cell population. To create pseudo-

bulk ATAC-seq peak data, we merged the peak sets for each cell population and resized the peaks to a width of 500 bp.”

REVIEWERS' COMMENTS:

Reviewer #1 (Remarks to the Author):

The authors fully addressed my concerns. Their results added in the revised version strengthened the conclusion that MER130 and MamRep434 contributed to the acquisition of cis-elements in putative MPCs.

Finally, please correct a typo in Figure 1b: "neuron-labeld cell"  "neuron-labeled cell"

Reviewer #2 (Remarks to the Author):

The authors responded to all my concerns in the revised manuscript. I don't have any major concerns. Here are some suggestions:

1. fonts in the figures are not consistent, and some of them are too small to read.
2. scRNA-seq data in the method section is just a Dropbox link, but the reference paper submitted their data to GEO. Please clearly note the data origin.